

# Quantifying the vertical transport of CHBr$_3$ and CH$_2$Br$_2$ over the Western Pacific

Robyn Butler[1], Paul I. Palmer[1], Liang Feng[1], Stephen J. Andrews[2], Elliot L. Atlas[3], Lucy J. Carpenter[2], Valeria Donets[3], Neil R. P. Harris[4], Stephen A. Montzka[5], Laura L. Pan[6], Ross J. Salawitch[7], and Sue M. Schauffler[6]

[1]School of GeoSciences, University of Edinburgh, UK
[2]Department of Chemistry, Wolfson Atmospheric Chemistry Laboratories, University of York, UK
[3]University of Miami, Florida, USA
[4]Department of Chemistry, University of Cambridge, UK
[5]National Oceanic and Atmospheric Administration, Boulder, USA
[6]National Center for Atmospheric Research, Boulder, Colorado, USA
[7]University of Maryland, College Park, Maryland, USA

*Correspondence to:* R. Butler (r.butler-2@sms.ed.ac.uk)

**Abstract.** We use the GEOS-Chem global 3-D atmospheric chemistry transport model to interpret atmospheric observations of bromoform (CHBr$_3$) and dibromomethane (CH$_2$Br$_2$) collected during the CAST and CONTRAST aircraft measurement campaigns over the Western Pacific, January–February, 2014. We use a new linearised, tagged version of CHBr$_3$ and CH$_2$Br$_2$, allowing us to study the influence of emissions from specific geographical regions on observed atmospheric variations. The

5    model describes 32%–37% of CHBr$_3$ observed variability and 15%-45% of CH$_2$Br$_2$ observed variability during CAST and CONTRAST, reflecting errors in vertical model transport. The model has a mean positive bias of 30% that is larger near the surface reflecting errors in the poorly constrained prior emission estimates. We find using the model that observed variability of CHBr$_3$ and CH$_2$Br$_2$ is driven by ocean emissions, particularly by the open ocean above which there is deep convection. We find that contributions from coastal oceans and terrestrial sources over the Western Pacific are significant above altitudes >6km, but

10   is still dominated by the open ocean emissions and by air masses transported over longer time lines than the campaign period. In the absence of reliable ocean emission estimates, we use a new physical age of air simulation to determine the relative abundance of halogens delivered by CHBr$_3$ and CH$_2$Br$_2$ to the tropical transition layer (TTL). We find that 6% (47%) of air masses with halogen released by the ocean reach the TTL within two (three) atmospheric e-folding lifetimes of CHBr$_3$ and almost all of them reach the TTL within one e-folding lifetime of CH$_2$Br$_2$. We find these gases are delivered to the TTL by a

15   small number of rapid convection events during the study period. Over the duration of CAST and CONTRAST and over our study region, oceans delivered a mean (range) CHBr$_3$ and CH$_2$Br$_2$ mole fraction of 0.46 (0.13–0.72) and 0.88 (0.71–1.01) pptv, respectively, to the TTL, and a mean (range) Br$_y$ mole fraction of 3.14 (1.81–4.18) pptv to the upper troposphere. Open ocean emissions are responsible for 75% of these values, with only 8% from coastal oceans.





## 1 Introduction

Halogenated very short-lived substances (VSLS) are gases that have a tropospheric e-folding lifetime of <6 months, which is shorter than the characteristic timescale associated with atmospheric transport of material from the surface to the tropopause. Natural sources of VSLS represent a progressively larger fraction of the inorganic halogen budget in the stratosphere that drives

halogen-catalysed ozone loss, as anthropogenic halogenated compounds continue to decline in accordance with international agreements. Quantifying the magnitude and variation of these natural VSLS fluxes to the stratosphere is therefore a research priority for environmental science. We focus on bromoform ($CHBr_3$) and dibromomethane ($CH_2Br_2$), which represent >80% of the organic bromine in the marine boundary layer and upper troposphere and are dominated by marine sources (WMO, 2007). We use aircraft observations of $CHBr_3$ and $CH_2Br_2$ collected over the western Pacific in January and February 2014 to

quantify the flux of these compounds to the stratosphere.

  The main sources of $CHBr_3$ and $CH_2Br_2$ include phytoplankton, particularly diatoms, and various species of seaweed (Carpenter and Liss, 2000; Carpenter et al., 2014; Quack and Wallace, 2003). The magnitude and distribution of these emissions reflect supersaturation of the compounds and nutrient-rich upwelling waters (Quack et al., 2007). Tropical, subtropical and shelf waters are important sources of $CHBr_3$ and $CH_2Br_2$ with high spatial and temporal variability (Ziska et al., 2013). Current

emission inventories, informed by sparse ship-borne data, have large uncertainties (Liang et al., 2010; Ordóñez et al., 2012; Warwick et al., 2006; Ziska et al., 2013; Hossaini et al., 2013; Tegtmeier et al., 2012). The atmospheric lifetime of $CHBr_3$ is ~24 days, determined primarily by photolysis and to a lesser extent by OH oxidation. $CH_2Br_2$ has an atmospheric lifetime of ~123 days determined by OH oxidation (Ko and Poulet, 2003).

  Ascent of $CHBr_3$ and $CH_2Br_2$ and their oxidation products to the upper troposphere lower stratosphere (UTLS) represent a

source of bromine that acts a catalysts for ozone loss in the stratosphere. Balloon-borne and satellite observations estimate that brominated VSLS and their degradation products contribute 2–8 ppt to stratospheric $Br_y$ (Dorf et al., 2008; McLinden et al., 2010; Salawitch et al., 2010; Sioris et al., 2006; Sinnhuber et al., 2002, 2005). Model estimates range between 2 and 7 ppt for this contribution (Aschmann et al., 2009; Fernandez et al., 2014; Hossaini et al., 2010; Liang et al., 2010; Ordóñez et al., 2012). This contribution mainly originates from areas of deep convection over the tropical Indian ocean, Western Pacific, and off the

Pacific coast of Mexico (Aschmann et al., 2009; Ashfold et al., 2012; Fueglistaler et al., 2004; Gettelman et al., 2002; Hossaini et al., 2012). The stratospheric community has categorized two methods of delivering VSLS to the stratosphere: 1) source gas injection (SGI), which describes the direct transport of the emitted halogenated compounds (e.g., $CHBr_3$ and $CH_2Br_2$); and 2) product gas injection (PGI), which refers to the transport of the degradation products of these emitted compounds. Previous model-based calculations (Aschmann and Sinnhuber, 2013; Tegtmeier et al., 2012; Liang et al., 2014; Hossaini et al., 2012)

have estimated that 15%-75% of the stratospheric inorganic bromine budget from VSLS is delivered by SGI, with uncertainty of the total $Br_y$ reflecting uncertainty of wet deposition of PGI product gases in the UTLS (Sinnhuber and Folkins, 2006; Liang et al., 2014).

  Active biological waters that coincide with regions of strong convection represent the major sources of VSLS to the upper troposphere (UT). The tropical tropopause layer (TTL) extends over a few kilometres and lies within this area of the UT





between the lapse rate minimum (∼12–13 km) and the cold point tropopause (∼17 km) (Gettelman and Forster, 2002). A slow transition between the thermodynamic structure of the convectively controlled troposphere to the radiatively controlled stratosphere gives it a smaller lapse rate than a saturated adiabatic up to the cold point (Fueglistaler et al., 2009; Pan et al., 2014; Zhou et al., 2004). This gives the TTL properties and structure of both the troposphere and the stratosphere, making it

the predominant transport pathway of SGI and PGI gases to the lower stratosphere. TTL temperatures vary zonally with the smallest values between 130°–180°E throughout the year. This region corresponds to the tropical warm pool over the Western Pacific, where convective activity is largest (Gettelman et al., 2002). Estimates of SGI within this region are highly dependent on the strength and spatial variability of source regions, and how they couple with atmospheric transport mechanisms.

We use the GEOS-Chem global 3-D chemistry to estimate the atmospheric flux of $CHBr_3$ and $CH_2Br_2$ to the TTL over the

western Pacific using measurements collected during two coincident airborne campaigns: the Coordinated Airborne Studies in the Tropics (CAST) and CONvective TRansport of Active Species in the Tropics (CONTRAST), January and February 2014 (Harris et al., 2016; Pan et al., 2016). In the next section we describe these two campaigns and the data used. Section 3 describes the GEOS-Chem model and how it is used to interpret the airborne data. In section 4 we evaluate the model and describe our results. The paper is concluded in section 5.

## 15  2  Observational data

### 2.1  CAST and CONTRAST $CHBr_3$ and $CH_2Br_2$ mole fraction data

We use $CHBr_3$ and $CH_2Br_2$ mole fractions from the CAST and CONTRAST aircraft campaigns (Harris et al., 2016; Pan et al., 2016). A comprehensive description of the data collection and analysis procedures used during the campaigns can be found in Andrews et al. (2016); here we provide only brief details of the $CHBr_3$ and $CH_2Br_2$ data.

Figure 1 shows the spatial distribution of whole air samples (WAS) collected during CAST and CONTRAST; other non-WAS halocarbon data, not analyzed here, have only recently become available. For CAST, WAS canisters were filled aboard the Facility for Airborne Atmospheric Measurements (FAAM) BAe-146 aircraft. These canisters were analysed for $CHBr_3$ and $CH_2Br_2$ and other trace compounds within 72 hours of collection. The WAS instrument was calibrated using the National Oceanic and Atmospheric Administration (NOAA) 2003 scale for $CHBr_3$ and the NOAA 2004 scale for $CH_2Br_2$. For CON-

TRAST, a similar WAS system was employed to collect $CHBr_3$ and $CH_2Br_2$ measurements on the NSF/NCAR Gulfstream-V HIAPER (High-performance Instrumented Airborne Platform for Environmental Research) aircraft. A working standard was used to regularly calibrate the samples, and the working standard was calibrated using a series of dilutions of high concentration standards that are linked to National Institute of Standards and Technology standards. The mean absolute percentage error for $CHBr_3$ and $CH_2Br_2$ measurements between 0−8 km is 7.7% and 2.2%, respectively, between the two WAS systems and

two accompanying GC/MS instruments.

Table 1 shows mean measurement statistics of $CHBr_3$ and $CH_2Br_2$ for the CAST and CONTRAST campaigns. $CHBr_3$ is generally more variable than $CH_2Br_2$ throughout the study region, reflecting its shorter atmospheric lifetime, so that sampling differences between CAST and CONTRAST will introduce larger differences for this gas. CAST measurements of $CHBr_3$





are typically lower than for CONTRAST, but CAST recorded the highest and lowest $CHBr_3$ mole fractions at $0-2$ km and $6-8$ km, respectively. We define the TTL from 13 km (Pan et al., 2014) to the local tropopause determined from the GMAO-FP analysed meteorological fields, as described below. CONTRAST measured a minimum $CHBr_3$ value indistinguishable from zero just below the TTL at 10–13 km. Measurements of $CH_2Br_2$ are generally consistent between CAST and CONTRAST at

all altitudes. There is only a small vertical gradient for $CH_2Br_2$ above 2 km with a mean value of $\sim$0.91 pptv. CONTRAST measured the lowest value of 0.21 pptv just below the TTL. Within the TTL, CONTRAST reports mean (maximum) values of 0.42 pptv (0.85 pptv) and 0.84 pptv (1.05 pptv) for $CHBr_3$ and $CH_2Br_2$, respectively, providing some evidence of rapid convection of surface emissions to the upper troposphere.

## 2.2 NOAA ground-based $CHBr_3$ $CH_2Br_2$ measurements

To evaluate our model of atmospheric $CHBr_3$ and $CH_2Br_2$, described below, we use independent surface measurements of these VSLS collected by the NOAA Earth System Research Laboratory (ERSL).

Table 2 shows the 14 geographical locations of the measurements we use, which are part of the ongoing NOAA/ESRL global monitoring program (http://www.esrl.noaa.gov/gmd). $CHBr_3$ and $CH_2Br_2$ measurements are obtained using WAS collected approximately weekly in paired steel flasks, which are then analysed by GC/MS. Further details about their sampling are given

in Montzka et al. (2011). In Appendix A, we evaluate the model using mean monthly statistics at each sites from 1st January 2005 to 31 December 2011.

## 3 The GEOS-Chem Global 3-D Atmospheric Chemistry Transport Model

To interpret CAST and CONTRAST data we use v9.02 of the GEOS-Chem global 3-D atmospheric chemistry transport model (www.geos-chem.org), driven by GMAO-FP analysed meteorological fields from the NASA Global Modelling and Assimila-

tion Office at NASA Goddard. For our experiments we degrade the meteorological analyses to a horizontal spatial resolution of $2°$ latitude $\times$ $2.5°$ longitude over 47 vertical levels. We describe below two new GEOS-Chem simulations that we developed to interpret observed variations of $CHBr_3$ and $CH_2Br_2$ during CAST and CONTRAST airborne campaigns: 1) a tagged simulation of $CHBr_3$ and $CH_2Br_2$ to better understand source attribution; and 2) an age of air simulation to improve understanding of the vertical transport of these short-lived halogenated compounds. For both simulations, we sample the model at the time

and location of CAST and CONTRAST observations.

## 3.1 Tagged $CHBr_3$ and $CH_2Br_2$ Simulation

The purpose of this simulation is to relate observed variations to surface emissions from individual sources and/or geographical regions. To achieve this we use pre-computed monthly 3-D fields of OH and photolysis rates for $CHBr_3$ and $CH_2Br_2$ from the full-chemistry version of GEOS-Chem, allowing us to linearise the chemistry so that we can isolate the contributions from

individual sources and geographical regions. The structure of the model framework follows closely other tagged simulations within GEOS-Chem (e.g., Jones et al. (2003); Palmer et al. (2003); Finch et al. (2014); Mackie et al. (2016)). We use the follow-





ing temperature $T$ dependent reaction rate constants that describe oxidation of $CHBr_3$ and $CH_2Br_2$ by the OH (Sander et al., 2011): for $CHBr_3$, $k(T) = 1.35 \times 10^{-12} \exp(-600/T) \, cm^3 molec^{-1}s^{-1}$; and for $CH_2Br_2$, $k(T) = 2.00 \times 10^{-12} \exp(-840/T)$ $cm^3 molec^{-1}s^{-1}$.

Figure 2 shows the land and (open and coastal) ocean tagged tracer regions we use in the GEOS-Chem model. Our geograph-5 ical definitions are informed by the NOAA ETOPO2v2 Global Relief map, which combines topography and ocean depth data at 2 minute spatial resolution: heights above 0 m are defined as land; -200 m < heights < 0 m are defined as coastal oceans; and heights below -200 m are open ocean. For tracers that spatially overlap we calculate their fractional contribution taking into account the area covered by land or ocean and local emission fluxes. We assign individual tracers to major islands within the study domain, including Guam (13.5°N, 144.8°E), Chuuk (7.5°N, 151.8°E), Palau (7.4°N, 134.5 °E) and Manus (2.1 °S, 10 147.4 °E). We assume these island land masses account for 100% of a grid box irrespective of whether their area fills the grid box. We have a total of 20 tagged tracers, evenly split between $CHBr_3$ and $CH_2Br_2$ including a total tracer and a background tracer.

Figure 3 shows the spatial distribution of $CHBr_3$ and $CH_2Br_2$ emissions that we use (Liang et al., 2010). These emissions are "top-down" estimates derived from airborne measurements in the troposphere and lower stratosphere in the Western Pacific 15 and North America. We use de-seasonalised monthly prior emissions from this inventory, with global annual totals of 425 Gg Br yr$^{-1}$ for $CHBr_3$ and 57 Gg Br yr$^{-1}$ for $CH_2Br_2$, and impose a monthly seasonal cycle at latitudes >30°N, following Parrella et al. (2012).

For model evaluation using the NOAA data, described above, we initialize the model in January 2004 with near-zero values until January 2013 with the first year discarded to minimize the impact of the initial conditions. For CAST/CONTRAST data, 20 we initialize the tagged tracers in January 2014 with near-zero values, and background initial conditions that were determined from a 12-month integration of the full-chemistry model that are subject only to atmospheric transport and loss processes. This approach minimizes any additional model error that has accumulated during the longer model integration. We sample at the time and location of each observation. For the NOAA data described above, we calculate monthly mean statistics from 1st January 2005–31 December 2011.

## 25 3.2 Physical age of air model calculation

We use the age of air simulation to understand the role and frequency of rapid convective systems to transport short-lived halogenated compounds to the TTL, in the absence of reliable bottom-up emission inventories.

We use the GEOS-Chem to determine the physical age of air $A$, building on previous studies (Finch et al., 2014), and use a consistent set of geographical regions used in our tagged $CHBr_3$ and $CH_2Br_2$ simulations as described above.

30 For each model tracer $(X_i)$ we define a surface boundary condition $B$ that linearly increases with time $t$ so that smaller values correspond to older physical ages: $B = f \times t \times R_x$, where $f$ is a scaling factor ($1 \times 10^{-15}$ s$^{-1}$), and $R_x$ denotes the fraction of the grid box relevant to a particular tagged tracer. We sample the resulting 3-D field of model tracers at the time and location of CAST and CONTRAST measurements. We initialise this model in July 2013 and run for 6 months until the start of January 2014 so that at least one e-folding lifetime of $CH_2Br_2$ has been achieved. The physical age of a tracer $i$ since





it first touched a land or ocean surface is calculated as $A_i = t_i - X_i/f$. By using the atmospheric transport model we take into account atmospheric dispersion. We do not take into account any chemical loss in this simulation.

To explicitly evaluate marine convection in GEOS-Chem we developed a short-lived tagged tracer with an e-folding lifetime of four days, comparable to that of methyl iodide ($CH_3I$) in the tropics (Carpenter et al., 2014). We emit the tracer with an equilibrium mole fraction of 1 pptv over all oceanic regions described in Figure 2. We initialise the model on 1st January 2014 with an empty 3-D atmospheric field and run for two months until 01/03/2014. Model output is archived every two hours and the model is sampled along the aircraft flight tracks. GEOS-Chem captures mean marine convective flow over the study region compared to $CH_3I$ observations. It also captures infrequent fast, large-scale convective transport with upper tropospheric ages of 3–5 days, but misses small-scale variations due to rapid convection. Results are explained in detail in Appendix B.

## 4 Results

### 4.1 Model Evaluation

We evaluate our tagged model of atmospheric $CHBr_3$ and $CH_2Br_2$ using NOAA surface data, and CAST and CONTRAST aircraft data during January and February 2014.

Evaluation using the NOAA data is described in Appendix A. In brief, the model generally has a positive bias but reproduces 30–60% of the seasonal variation, depending on geographical location. Model errors in reproducing the observed seasonal cycle reflect errors in production and loss rates. The model generally has less skill at reproducing observations collected at coastal sites close to emission sources.

Figure 4 compares modelled and observed $CHBr_3$ and $CH_2Br_2$. It shows that GEOS-Chem has a 30% percentage bias for both gases during CAST and CONTRAST, which we calculate using: $100/N_i \sum_i (model_i - obs_i)/(max(mod_i, obs_i))$. We assume this bias is due primarily to errors in prior emission and remove it from subsequent calculations. We find that the model can reproduce more than 30% of the observed variability of $CHBr_3$ from CAST and CONTRAST and between 15% (CAST) and 45% (CONTRAST) of the observed variability of $CH_2Br_2$. In general, GEOS-Chem has poorer skill at reproducing observed near-surface variations, reflecting errors in prior emissions. We find that the frequency distribution of the model minus observation residuals are similar for CAST and CONTRAST but with an offset from zero, reflecting a systematic error that is likely due to an error in prior emissions. Differences between CAST and CONTRAST reflect the bias towards boundary-layer sampling during CAST where measurements are more sensitive to fresh surface emissions. There are smaller differences between CAST and CONTRAST for $CH_2Br_2$. It has a longer chemical lifetime making it comparable with the mean mole fraction associated with the troposphere.

### 4.2 Tagged-VSLS model output

Figures 5 (Figure 6) show mean land, and open and coastal ocean tagged $CHBr_3$ ($CH_2Br_2$) tracers over altitude compared to model convective mass flux. There is a strong region of convection south of Chuuk and along the equator that transports gases



directly from open oceanic emission sources to the mid-troposphere. Above the mid-troposphere (10 km) the mean convective mass fluxes get smaller with gases being dissipated progressively more in the horizontal. As we show below this leads to an inverted 'S' shape in the vertical profiles of $CHBr_3$ and $CH_2Br_2$, which is observed by both campaigns (Harris et al., 2016; Pan et al., 2016). There is also a strong convection region west of Papua New Guinea/north of Australia, close to land and coastal

emissions.

The model shows $CHBr_3$ mean mole fractions of 2 ppt throughout the boundary layer (0–2 km) emitted from open ocean emissions, but deplete quickly over the vertical profile due to their short atmospheric lifetime. Throughout the TTL, mean $CHBr_3$ mole fractions range 0.2–0.6 ppt over the study domain and campaign period mainly due to open ocean emissions. Coastal emissions are typically much larger than open ocean emissions but they play a much smaller role in observed variations

throughout the troposphere despite coinciding with the strong convective regions over Papua New Guinea/north of Australia. Prevailing easterly transport of gases over the region is dominated by the vast area of open ocean sources that appear to weaken the magnitude of spatially limited coastal emissions (Andrews et al., 2016; Pan et al., 2016). Averaged over the campaign, coastal and terrestrial sources of $CHBr_3$ show little influence above 6 km. The vertical and spatial distribution of $CH_2Br_2$ mole fractions is consistent with $CHBr_3$ although they deplete less rapidly with altitude by virtue of their longer atmospheric lifetime.

At the TTL, averaged over the campaign study, $CH_2Br_2$ mole fractions range 0.1–0.3 ppt mainly due to smaller magnitude of ocean emissions compared to $CHBr_3$. Coastal and terrestrial sources contribute up to 0.1 ppt of $CH_2Br_2$ in the TTL.

Figure 7 shows percentage contributions of geographical tracers to CAST and CONTRAST $CHBr_3$ observations. Ocean emissions provide the largest fractional contribution to $CHBr_3$ during CAST, typically more than 70% throughout the low to mid troposphere of which 60–80% is from the open ocean. Coastal ocean and land emissions represent a much smaller

contribution to $CHBr_3$ at lower altitudes, but increase their influence above 6 km in the CONTRAST data with contributions from geographical regions immediately outside the study region reaching a maximum of 20% of the total $CHBr_3$ tracer in the TTL. This is represented in the inverted 'S' shape observed over the vertical profile, as mentioned above. This reflects deep convection of air masses over the region, which has only a small amount of detrainment in the mid-troposphere followed by advection of these air masses in the upper troposphere. Island land masses generally represent a minor contribution through

the vertical profile and we have excluded them from further analysis.

Figure 8 shows the same as Figure 7 but for $CH_2Br_2$. The largest contributions to total $CH_2Br_2$ over the campaign period are from the oceans, particularly from the open ocean. They typically represent 20% of the total $CH_2Br_2$ and reaching a maximum of 34% in the TTL for the CONTRAST measurements. Maximum contributions of coastal emission sources peak at 5% of total $CH_2Br_2$ tracer in the TTL, much less than for $CHBr_3$. The remaining contributions are representative of emissions before

the campaign period. The longer lifetime of $CH_2Br_2$ mean that these mole fractions have a greater influence over the campaign profile compared to $CHBr_3$.

### 4.3   Physical Age of Air

Figure 9 shows how the probability density of the age of air, $A$, from different geographical tracers changes over altitude. The age of air has a bi-modal distribution peaking at approximately 60 days and 200 days. The younger age distribution is





dominated by air lofted over the open ocean, while the older age distribution is dominated by air lofted over the land and coastal oceans. At progressively higher altitude regions the distributions generally age, as expected, but above the boundary layer the median values of both modes remain relatively constant at approximately 70 days and 210 days, respectively. The older peak is representative of emission sources have been imported into the study region from outside the study domain.

Contributions from coastal ocean and terrestrial emissions are small compared to the contribution from the larger area of open ocean emissions. The oldest ages, which approach the time of the study period, reflect the accumulation of near-zero mole fractions. At higher altitudes we find that the probability distributions become less smooth, reflecting more variation in ages. At these altitudes we also find a progressively larger (but still minor) contribution from coastal emissions. We using our $CH_3I$-like tracer that air masses can be transported to the TTL within 3–5 days but these are infrequent events (Appendix B).

Assuming an indicative e-folding atmospheric lifetime $\tau$ of 24 days for $CHBr_3$ and 123 days for $CH_2Br_2$ we find that the majority of air lofted over the ocean has an age within $3\tau_{CHBr_3}$ and $1\tau_{CH_2Br_2}$. We find that 11% (52%) of oceanic emissions reach the TTL within $2\tau_{CHBr_3}$ ($3\tau_{CHBr_3}$) of which 53.6% (34.9%) are from the open oceans. Contributions from the land and coastal oceans are negligible as they dominate the older age profile throughout the vertical. Coastal based emissions show some influence at higher altitudes due to the difference in entire ocean and open ocean age profiles, but they have no strength

as an individual geographical tracer. The corresponding statistics for $CH_2Br_2$ are 95% of air lofted over the ocean reaches the TTL within $1\tau_{CH_2Br_2}$ of which 88% is lofted over the open ocean.

Figure 10 is the same as Figure 9 but sampled along CAST and CONTRAST flight tracks. The atmospheric sampling adopted by the CAST and CONTRAST campaigns capture the bi-modal distribution of physical ages discussed above. Despite intensive measurements around coastal land masses of the region, CAST did not very well capture coastal emissions. We also

find that CONTRAST generally better samples both modes of the distribution, reflecting the more extensive horizontal and vertical sampling domain and the larger number of collected measurements.

Figure 11 is the distribution of oceanic $CHBr_3$ mixing ratios in the troposphere, in all systems and in only the highest convective systems compared with associated age. Throughout the troposphere, $CHBr_3$ mole fractions generally decrease with age. The only exception is at the near-surface where land emissions dominate the older age profile. Within the TTL, the

highest $CHBr_3$ mole fractions are associated with the youngest age of air (24–48 days), but this represents only 5% of the air transported to the TTL. The peak frequency for the mean age of air is 48–72 days, corresponding to $3\tau_{CHBr_3}$ and median values of 0.5 pptv $CHBr_3$ from oceanic emission sources. The highest values of model convective mass flux do not account for all the high $CHBr_3$ mole fractions within the TTL. Less than 0.5% (2%) of air being transported to the TTL within 20–40 (48–72) days of emission are associated with high convection events. Weaker, mean convection plays an important role in more

consistently transporting large mole fractions to the free troposphere that is then transported more slowly to the TTL.

To estimate the mean observed transport of $CHBr_3$ and $CH_2Br_2$ to the TTL we remove the calculated model bias as described above in section 4.1. Figure 12 shows the resulting corrected mean vertical profiles. We calculate the uncertainties using the upper and lower limits of the bias correction, which are based on $CHBr_3$ and $CH_2Br_2$ data that are $\pm2$ mean absolute deviations from the observed mean mole fractions. For $CHBr_3$ and $CH_2Br_2$ we find biases that range -8%–80% and 19%–

43%, respectively, which we then apply to the model values throughout the atmosphere over the campaign period. We find





that resulting mean model values underestimate observed $CHBr_3$ and $CH_2Br_2$ between 9–12 km, above the main region of convective outflow, with the observations inside the model uncertainty with the exception of $CH_2Br_2$. Mean model values within the TTL (above 13 km and below the local tropopause) reproduce mean observations. Based on this bias correction approach we infer a mean mole fraction and range of 0.46 (0.13–0.72) ppt and 0.88 (0.71–1.01) ppt of $CHBr_3$ and $CH_2Br_2$

being transported to the TTL during January and February, 2014. This is consistent with a contribution of 3.14 (1.81–4.18) pptv of Br to the TTL $Br_y$ budget over the region of the campaign. This is consistent with Navarro et al. (2015) which estimates VSLS contribution over the Pacific from observations in 2013 and 2014. It estimates 3.27±0.47 pptv of bromine from $CHBr_3$, $CH_2Br_2$ and other minor VSLS sources at the tropopause level (17 km).

Based on average observed surface values of $CHBr_3$ (1.13 ppt) and $CH_2Br_2$ (1.02 ppt) over the campaign we infer that 40%

and 86% of these emitted gases, respectively, are directly injected into the TTL over our study domain. The larger percentage for $CH_2Br_2$ is consistent with its longer lifetime. Our value of 40% for the $CHBr_3$ SGI falls within previously reported values that range 15%–76% (median $\simeq$50%) (Sinnhuber and Folkins, 2006; Hossaini et al., 2010; Tegtmeier et al., 2012; Aschmann and Sinnhuber, 2013), but is lower than the associated median value. One possible reason for the negative bias in our SGI estimate for $CHBr_3$ is the bias correction approach we adopted for our analysis. Our bias correction is simple but not does take

account for vertical variations in atmospheric transport. We calculated a mean atmospheric bias, but clearly the model bias is much larger at lower altitudes, reflecting errors in emission estimates. However, we find that model bias at altitudes >10 km (29%) is comparable to the bias calculated using all data (31%). There is much less vertical variation in bias for $CH_2Br_2$ because of its longer atmospheric lifetime.

## 5   Discussion and Concluding Remarks

We used the GEOS-Chem chemistry transport model to interpret mole fraction measurements of $CHBr_3$ and $CH_2Br_2$ over the Western Pacific during the CAST and CONTRAST campaigns, January–February 2014. We found that the model reproduced 30% of $CHBr_3$ measurements and 15% (45%) CAST (CONTRAST) $CH_2Br_2$, but had a mean positive bias of 30% for both compounds. CAST mainly sampled the marine boundary layer (70% of observations) so that biases in prior surface emissions have a greater influence on CAST than CONTRAST, which sampled throughout the troposphere.

To interpret the CAST and CONTRAST measurements of $CHBr_3$ and $CH_2Br_2$ we developed two new GEOS-Chem model simulations: 1) a linearised tagged simulation so that we could attribute observed changes to individual sources and geographical regions, and 2) an age of air simulation to improve understanding of the vertical transport of these compounds, acknowledging that more conventional photochemical clocks are difficult to use without more accurate boundary conditions provided by surface emission inventories.

We have three main conclusions. First, we found that open ocean emissions of $CHBr_3$ and $CH_2Br_2$ are primarily responsible for observed atmospheric mole fractions of these gases over the Western Pacific. Emissions from open ocean sources represent up to 75% of total $CHBr_3$, with the largest fractional contribution in the lower troposphere. Coastal ocean and terrestrial sources typically contribute 7% to total atmospheric $CHBr_3$ but reach a maximum of 20% in the TTL due to advection of air masses





convected from areas outside the study region. Based on this model interpretation, we infer that CAST observations of $CHBr_3$, which are mainly in the lower troposphere, are dominated by open ocean sources. In contrast, CONTRAST measurements have a mix of sources, including a progressively larger contribution from coastal ocean and terrestrial sources in the upper troposphere. Tropospheric measurements of $CH_2Br_2$, which has a longer atmospheric lifetime than $CHBr_3$, are dominated by

sources from before the campaign. The open ocean source typically represents only 20% of the observed variations of $CH_2Br_2$ emitted during the campaign region throughout the troposphere.

Second, using our age of air simulation, we find that the highest $CHBr_3$ and $CH_2Br_2$ mole fractions in the TTL correspond to the youngest air masses being transported from oceanic sources, predominantly the open ocean. Within the TTL, the highest $CHBr_3$ mole fractions are associated with the youngest age of air (24–48 days), but this represents only 5% of the air transported

to the TTL. Weaker, slower convection processes are responsible for consistently transporting higher mole fractions to the UT and TTL. The majority of air (40%) is being transported to the TTL is within $3\tau_{CHBr_3}$ (48–72 days) corresponding to lower mole fractions and the majority of weaker convection events.

And third, we estimated the flux of $CHBr_3$ and $CH_2Br_2$ to the TTL using model data that have been corrected for bias. We calculated a mean and range of values 0.46 (0.13–0.72) pptv and 0.88 (0.71–1.01) pptv for $CHBr_3$ and $CH_2Br_2$, respectively,

which represent 40% and 86% of estimated surface emissions. Together, they correspond to a total of 3.14 (1.81–4.18) pptv Br to the TTL. Our flux estimate for $CHBr_3$ is lower than previous studies that have reported values closer to 50%.

*Acknowledgements.* R.B. and P.I.P. designed the computation experiments and R.B. conducted the experiment with contributions from L.F. about the tagged model. R.B. and P.I.P. wrote the manuscript. We are grateful to the Harvard University GEOS-Chem group who maintain the model. R.B. was funded by the United Kingdom Natural Environmental Research Council (NERC) studentship NE/1528818/1, L.F.

was funded by NERC grant NE/J006203/1, and P.I.P. gratefully acknowledges his Royal Society Wolfson Research Merit Award. R.S. acknowledges support from the U.S. National Science Foundation (NSF). E.A. acknowledges support from NSF Grant AGS1261689 and thanks R. Lueb, R. Hendershot, X. Zhu, M. Navarro, and L. Pope for technical and engineering support. CAST is funded by NERC and STFC, with grants NE/ I030054/1 (lead award), NE/J006262/1, 472 NE/J006238/1, NE/J006181/1, NE/J006211/1, NE/J006061/1, NE/J006157/1, NE/J006203/1, NE/J00619X/1 (UoYork CAST measurements), and NE/J006173/1. The CONTRAST experiment is sponsored by the NSF.

CONTRAST data are publicly available for all researchers and can be obtained at http://data.eol.ucar.edu/master_list/?project=CONTRAST. The NOAA surface data is available at http://www.esrl.noaa.gov/gmd/dv/ftpdata.html.




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



**Table 1.** Mean measurement statistics for $CHBr_3$ and $CH_2Br_2$ mole fraction data as a function of altitude for CAST and CONTRAST aircraft campaigns. $\overline{x}$, $\sigma$, and $n$ denote the mean value, the standard deviation, and the number of data points used to determine the statistics.

| Altitude | $CHBr_3$ | | $CH_2Br_2$ | |
| --- | --- | --- | --- | --- |
| | CAST | CONTRAST | CAST | CONTRAST |
| km | $\overline{x}$, $1\sigma$, & range [ppt]; $n$ | $\overline{x}$, $1\sigma$, & range [ppt]; $n$ | $\overline{x}$, $1\sigma$, & range [ppt] | $\overline{x}$, $1\sigma$, & range [ppt] |
| $0-2$ | 0.95, 0.45, 0.42$-$3.00; 502 | 0.89, 0.23, 0.51$-$1.55; 75 | 1.01, 0.13, 0.72$-$1.64 | 1.07, 0.11, 0.83$-$1.27 |
| $2-4$ | 0.61, 0.16, 0.29$-$0.98; 147 | 0.62, 0.18, 0.29$-$1.24; 48 | 0.91, 0.05, 0.73$-$1.06 | 0.94, 0.09, 0.78$-$1.13 |
| $4-6$ | 0.44, 0.17, 0.03$-$0.79; 59 | 0.56, 0.18, 0.20$-$1.12; 65 | 0.88, 0.06, 0.69$-$1.00 | 0.91, 0.09, 0.73$-$1.12 |
| $6-8$ | 0.38, 0.25, 0.02$-$0.81, 53 | 0.60, 0.20, 0.24$-$1.01; 43 | 0.85, 0.11, 0.63$-$1.06 | 0.90, 0.10, 0.70$-$1.06 |
| $8-10$ | 0.48, 0.34, 0.14$-$0.82; 2 | 0.62, 0.17, 0.24$-$1.00; 43 | 0.90, 0.13, 0.77$-$1.03 | 0.93, 0.09, 0.72$-$1.07 |
| $10-13$ | – | 0.59, 0.25, 0.00$-$1.38; 130 | – | 0.87, 0.19, 0.21$-$1.10 |
| TTL | – | 0.48, 0.16, 0.18$-$1.17; 280 | – | 0.86, 0.08, 0.64$-$1.06 |



**Table 2.** Location and code of NOAA/ESRL ground-based stations. All located at the surface with exceptions of SUM (3210 m), MLO (3397 m) and SPO (2810 m).

| Station | Name | Lat | Lon |
|---------|------|-----|-----|
| ALT | Alert, NW Territories, Canada | 82.5°N | 62.3°W |
| SUM | Summit, Greenland | 72.6°N | 38.4°W |
| BRW | Pt. Barrow, Alaska, USA | 71.3°N | 156.6°W |
| MHD | Mace Head, Ireland | 53.0°N | 10.0°W |
| LEF | Wisconsin, USA | 45.6°N | 90.2°W |
| HFM | Massachusetts, USA | 42.5°N | 72.2°W |
| THD | Trinidad Head, USA | 41.0°N | 124.0°W |
| NWR | Niwot Ridge, Colorado, USA | 40.1°N | 105.6°W |
| KUM | Cape Kumukahi, Hawaii, USA | 19.5°N | 154.8°W |
| MLO | Mauna Loa, Hawaii, USA | 19.5°N | 155.6°W |
| SMO | Cape Matatula, American Samoa | 14.3°S | 170.6°W |
| CGO | Cape Grim, Tasmania, Australia | 40.7°S | 177.8°E |
| PSA | Palmer Station, Antarctica | 64.6°S | 64.0°W |
| SPO | South Pole | 90.0°N | – |





**Table 3.** Seasonal break down of statistics for NOAA ground station sites (Table 2) showing $r^2$ correlations between observed and climatological monthly mean $CHBr_3$ and $CH_2Br_2$ mole fraction data and corresponding model bias values.

| | CHBr$_3$ | | | | CH$_2$Br$_2$ | | | |
| --- | --- | --- | --- | --- | --- | --- | --- | --- |
| | DJF | MAM | JJA | SON | DJF | MAM | JJA | SON |
| Station | $r^2$, %bias | $r^2$, %bias | $r^2$, %bias | $r^2$, %bias | $r^2$, %bias | $r^2$, %bias | $r^2$, %bias | $r^2$, %bias |
| ALT | 0.00, 3.8 | 0.55, 0.1 | 0.05, 5.5 | 0.43, 19.3 | 0.09, 12.4 | 0.21, 0.0 | 0.23, 10.0 | 0.31, 21.0 |
| SUM | 0.05, 25.1 | 0.01, −17.1 | 0.23, −12.0 | 0.54, 20.6 | 0.06, −2.7 | 0.15, −13.0 | 0.15, 4.8 | 0.60, 7.0 |
| BRW | 0.00, −41.3 | 0.52, −30.2 | 0.13, −26.5 | 0.80, −26.5 | 0.15, 9.5 | 0.07, −8.7 | 0.00, −4.4 | 0.14, 15.9 |
| MHD | 0.00, −40.8 | 0.18, −72.4 | 0.04, −80.6 | 0.08, −61.2 | 0.05, −20.5 | 0.11, −35.9 | 0.14, −42.7 | 0.03, −16.8 |
| LEF | 0.03, 45.7 | 0.01, 15.5 | 0.03, 39.3 | 0.73, 51.2 | 0.17, 13.4 | 0.25, 3.1 | 0.44, 18.2 | 0.66, 20.8 |
| HFM | 0.06, 52.2 | 0.01, 30.1 | 0.15, 46.9 | 0.38, 52.3 | 0.03, 20.0 | 0.06, 9.3 | 0.22, 27.1 | 0.45, 25.7 |
| THD | 0.19, 55.3 | 0.15, 15.8 | 0.36, 11.7 | 0.17, 40.2 | 0.09, 16.9 | 0.06, 0.9 | 0.01, 8.6 | 0.06, 18.0 |
| NWR | 0.02, 43.3 | 0.49, 25.9 | 0.01, 21.8 | 0.31, 38.7 | 0.20, 2.3 | 0.55, 4.9 | 0.40, 11.9 | 0.55, 14.7 |
| KUM | 0.00, 20.2 | 0.37, −1.9 | 0.05, 0.9 | 0.01, 6.8 | 0.25, −0.3 | 0.50, 4.4 | 0.47, 15.6 | 0.38, 9.9 |
| MLO | 0.18, 61.9 | 0.60, 60.3 | 0.02, 65.1 | 0.58, 64.7 | 0.14, 14.8 | 0.32, 15.2 | 0.21, 22.8 | 0.27, 25.0 |
| SMO | 0.23, 8.2 | 0.02, −4.9 | 0.04, 3.0 | 0.11, 4.7 | 0.39, 6.9 | 0.38, −0.9 | 0.19, −0.2 | 0.09, 5.6 |
| CGO | 0.23, −39.0 | 0.01, −12.8 | 0.05, 7.7 | 0.00, −19.6 | 0.13, −8.7 | 0.13, −1.6 | 0.04, −1.4 | 0.12, −9..5 |
| PSA | 0.19, −13.9 | 0.25, 26.4 | 0.01, 31.7 | 0.05, −1.9 | 0.00, −1.7 | 0.29, 11.7 | 0.15, 10.3 | 0.05, −2.0 |
| SPO | 0.50, 6.6 | 0.12, 6.7 | 0.07, 19.2 | 0.11, −7.3 | 0.01, 4.8 | 0.06, 4.6 | 0.11, 3.7 | 0.00, −0.6 |

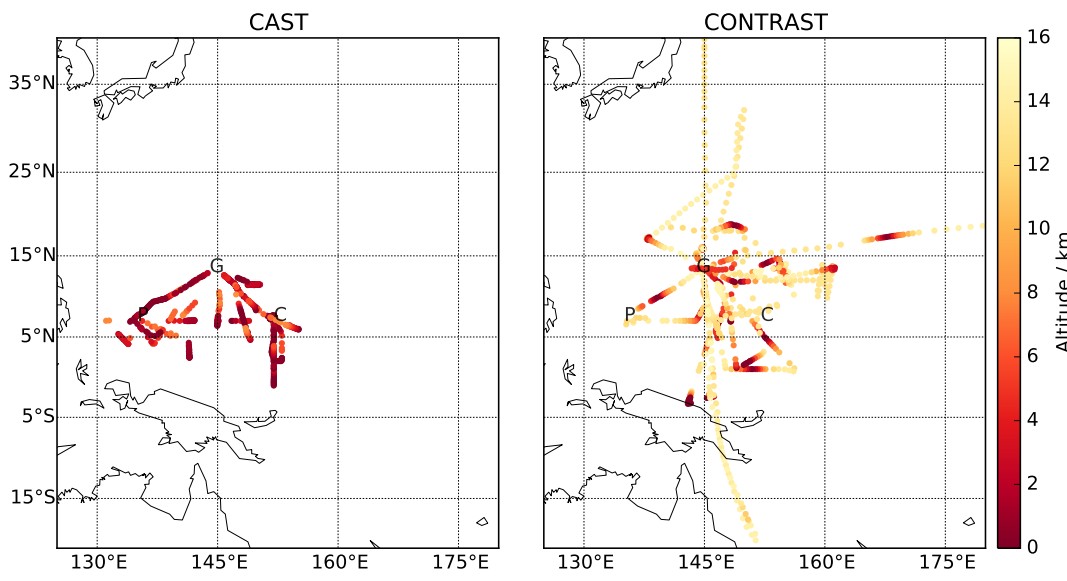

**Figure 1.** Distribution of measurements of $CHBr_3$ and $CH_2Br_2$ from the CAST (left) and CONTRAST (right) aircraft campaigns as a function of altitude (km). Relevant island waypoints are shown inset: Guam (G), Palau (P), and Chuuk (C).





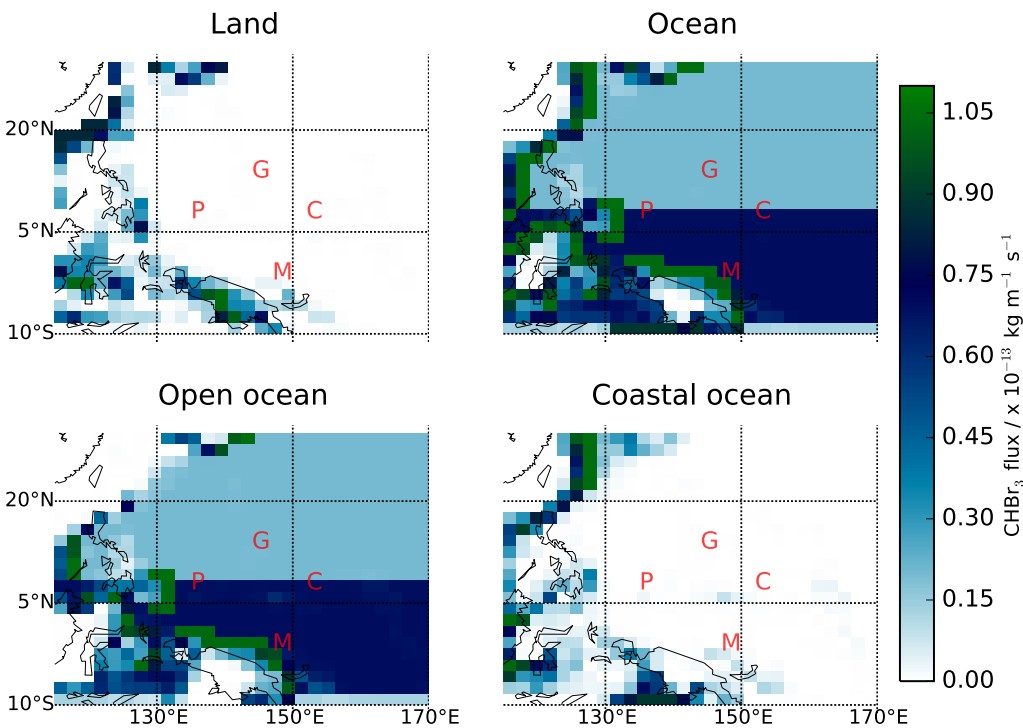

**Figure 2.** Flux of CHBr$_3$ from land and ocean, the latter separated in to its open and coastal ocean tracers. Guam (G), Palau (P), Chuuk (C) and Manus (M).





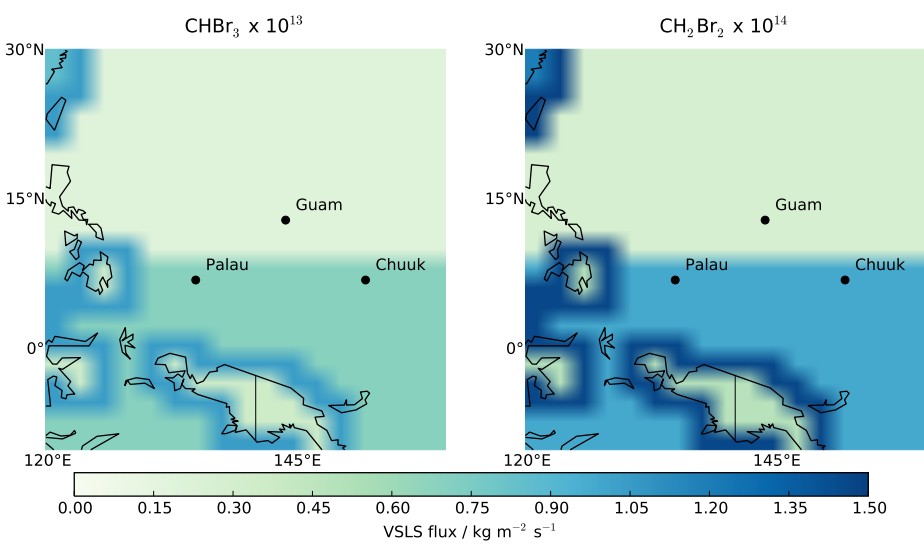

**Figure 3.** Surface emissions of CHBr$_3$ ($10^{13}$kg/m$^2$/s) and CH$_2$Br$_2$ ($10^{14}$kg/m$^2$/s) taken from Liang et al. (2010) for the time and region of the CAST and CONTRAST campaigns.



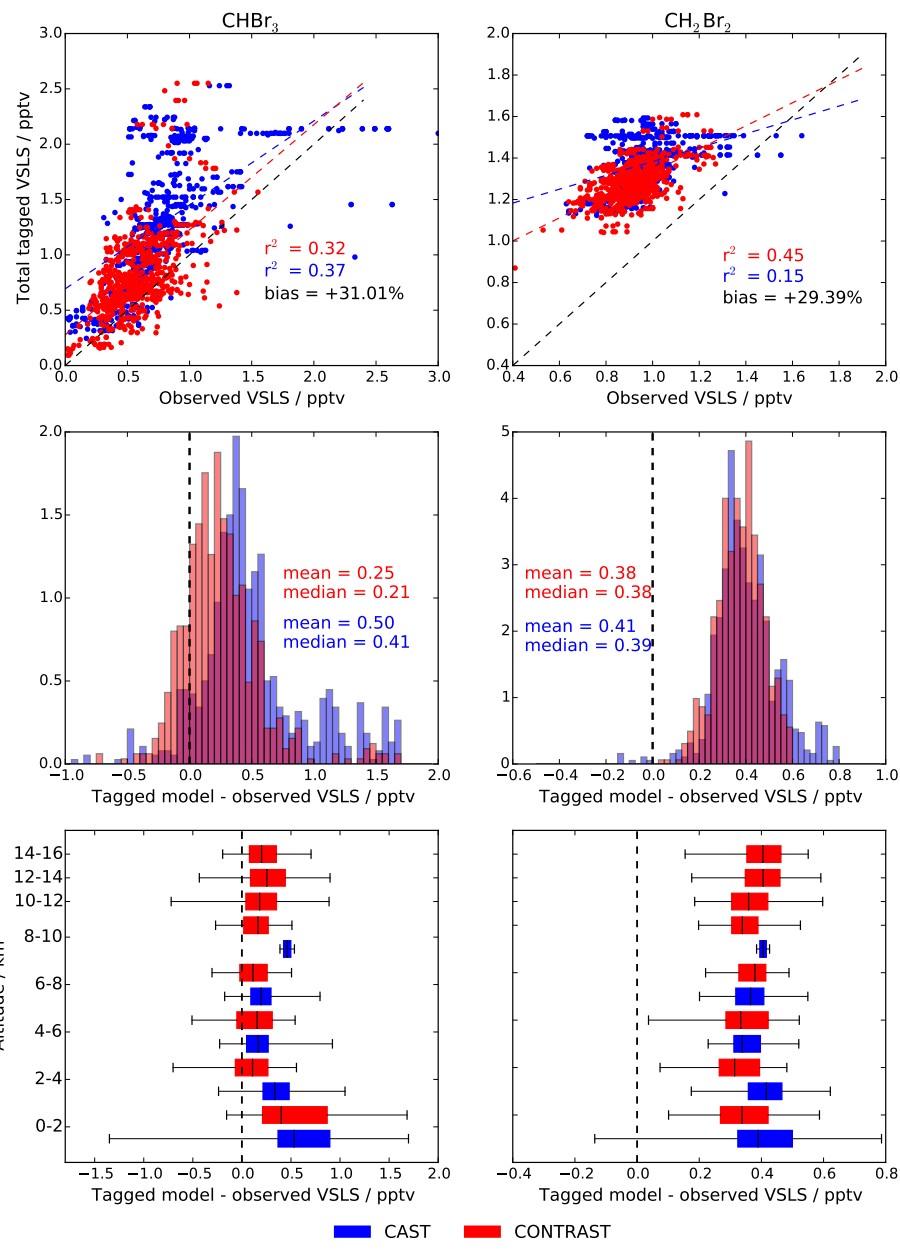

**Figure 4.** Comparisons of model and observed (left) $CHBr_3$ and (right) $CH_2Br_2$ mole fractions from the (blue) CAST and (red) CONTRAST aircraft campaigns. The top, middle and bottom rows display the comparison as a scatter plot (with the Pearson correlation, linear best-fit line, and percentage bias inset); a frequency distribution of the model minus observed values; and a vertical profile of box-and-whiskers of the model minus observed values.

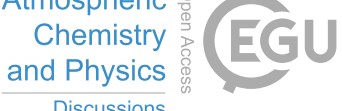

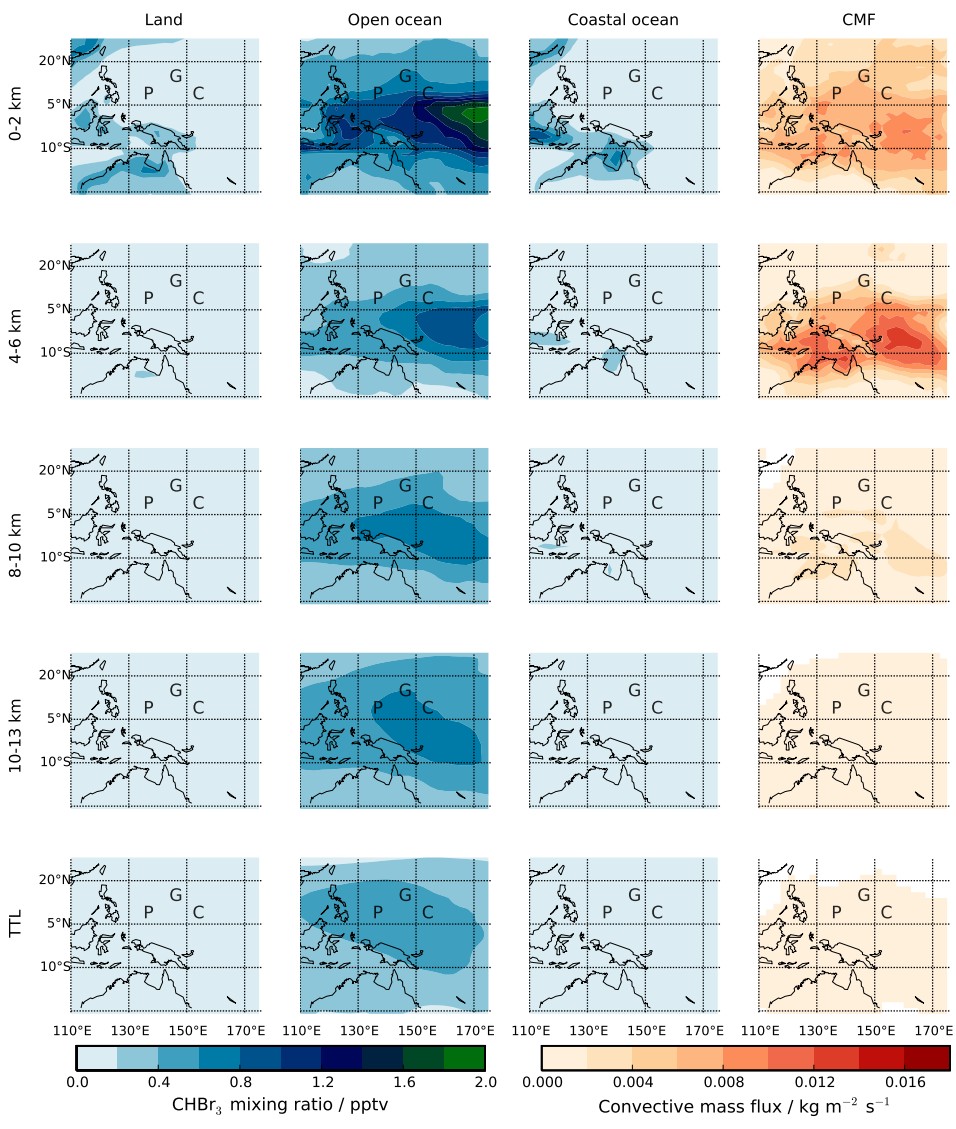

**Figure 5.** Model mole fractions (pptv) of CHBr$_3$ over the study domain as a function of altitude, averaged between 18/01/14 and 28/02/14, from the land (column 1), open ocean (column 2) and coastal ocean (column 3) tagged tracers. The corresponding mean model convective mass flux (kg/m$^2$/s) is shown in column 4. Tagged tracers are averaged from 2-hour fields and convective mass fluxes are averaged from daily fields.





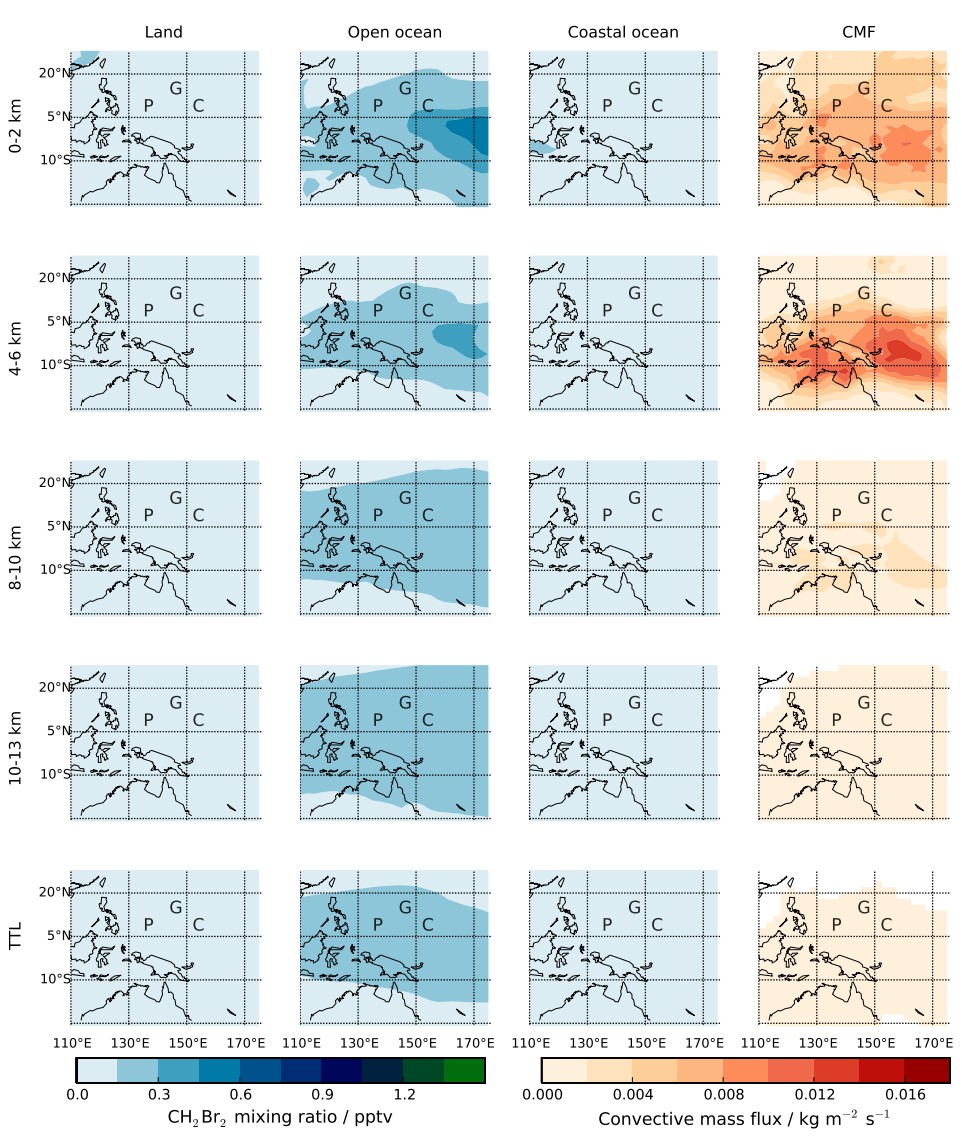

**Figure 6.** As Figure 5 but for CH$_2$Br$_2$.





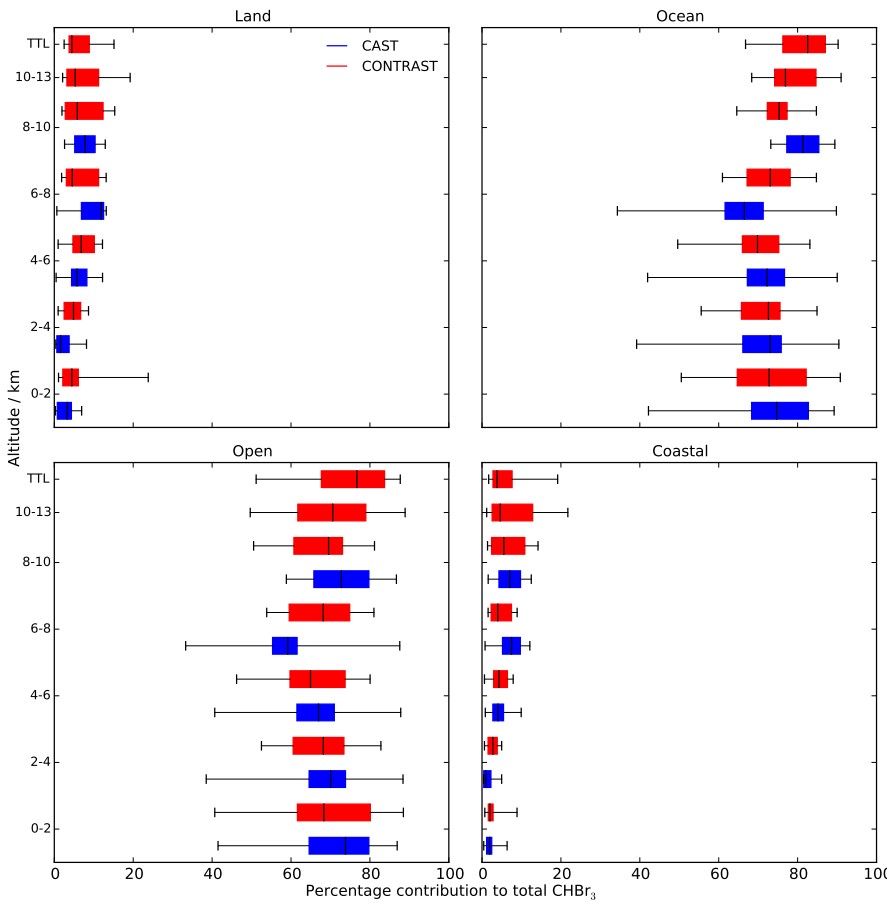

**Figure 7.** The percentage contributions, expressed as a box and whiskers plot, from land and ocean sources to total CHBr$_3$ sampled through-out the troposphere during (blue) CAST and (red) CONTRAST campaigns.



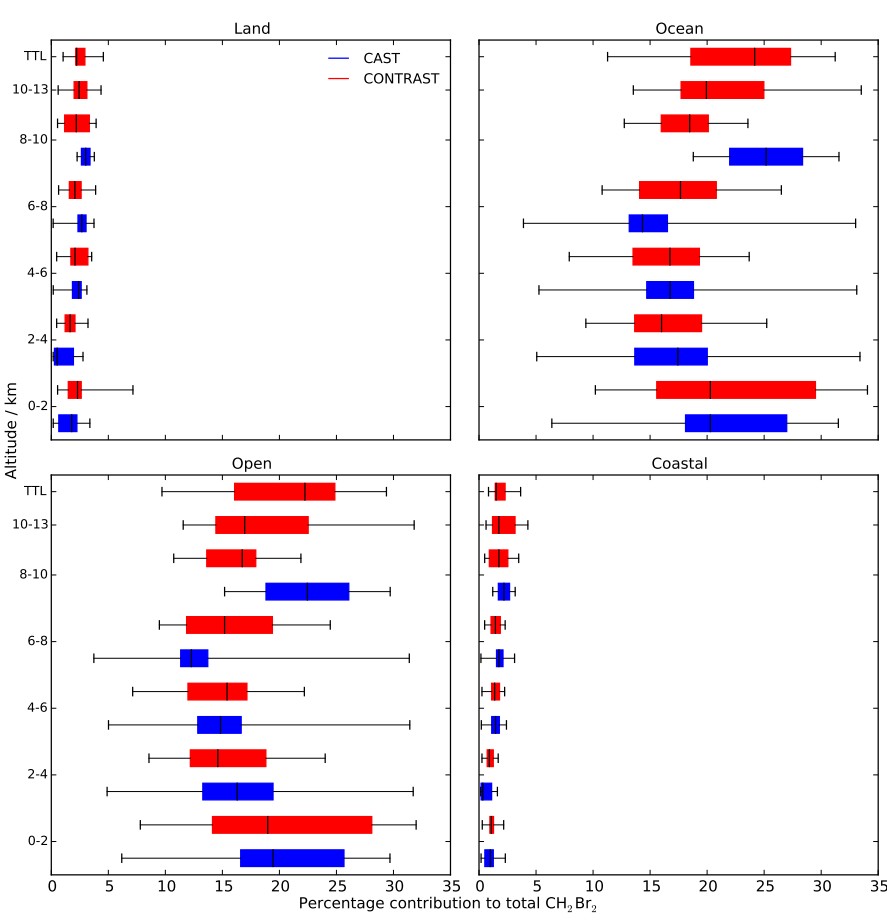

**Figure 8.** As Figure 7 but for $CH_2Br_2$.





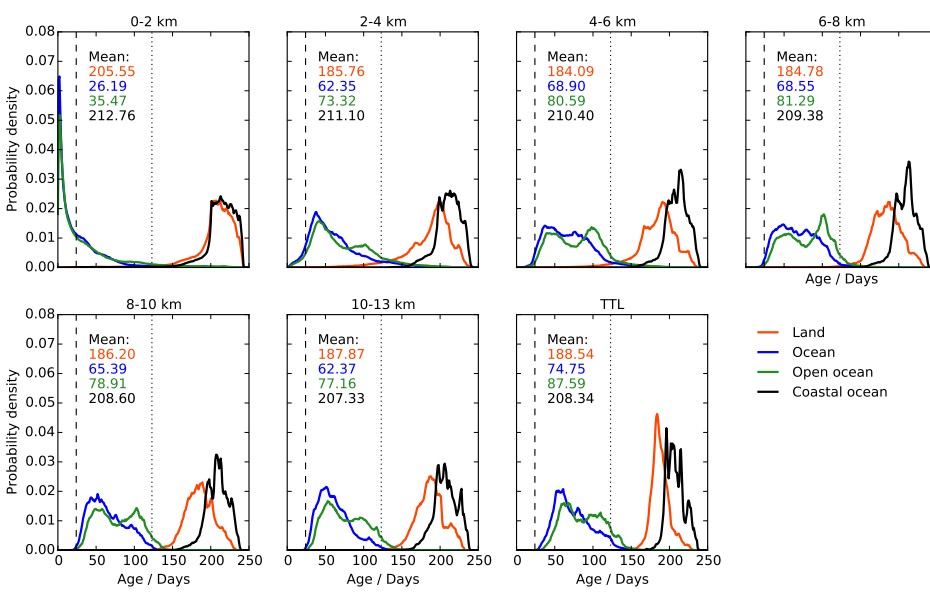

**Figure 9.** Frequency distributions of probability density function of the age of air *A* for (orange) land, (blue) entire ocean, and (green) open and (black) coastal ocean tracers in 2 km altitude regions up to the TTL (13 km to the tropopause) averaged over 18/01/2014–28/02/2014.





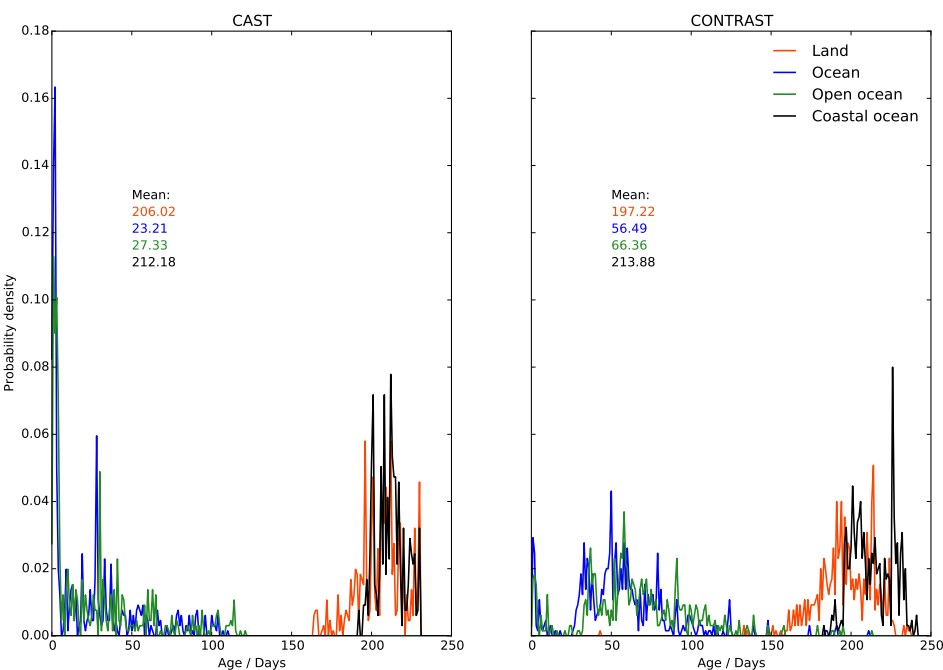

**Figure 10.** As Figure 9 but sampled along CAST and CONTRAST flight tracks.





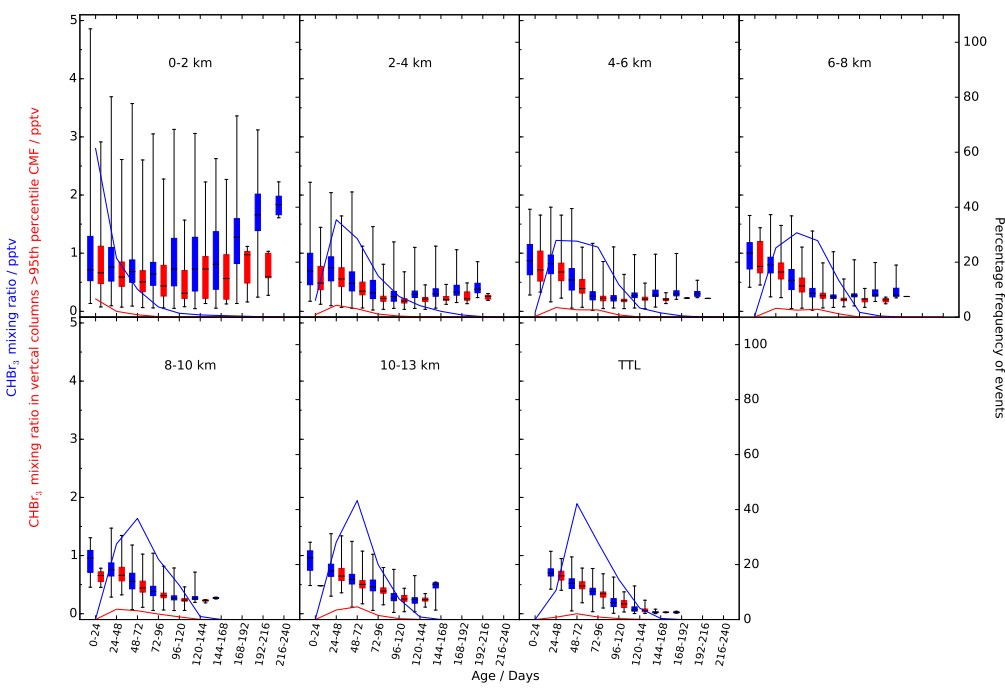

**Figure 11.** Box and whiskers plot of model CHBr$_3$ mole fractions from the entire ocean tracer as a function of 2 km altitude intervals and nominal 24-day e-folding lifetime. Data are averaged over 18/01/2014–28/02/2014 and over 10°S-30°N, 125-175°E. All data are shown in blue and data corresponding to convective mass fluxes >95th percentile are show in red. Solid lines denotes the percentage of occurrence rate over the period and region denoted above.





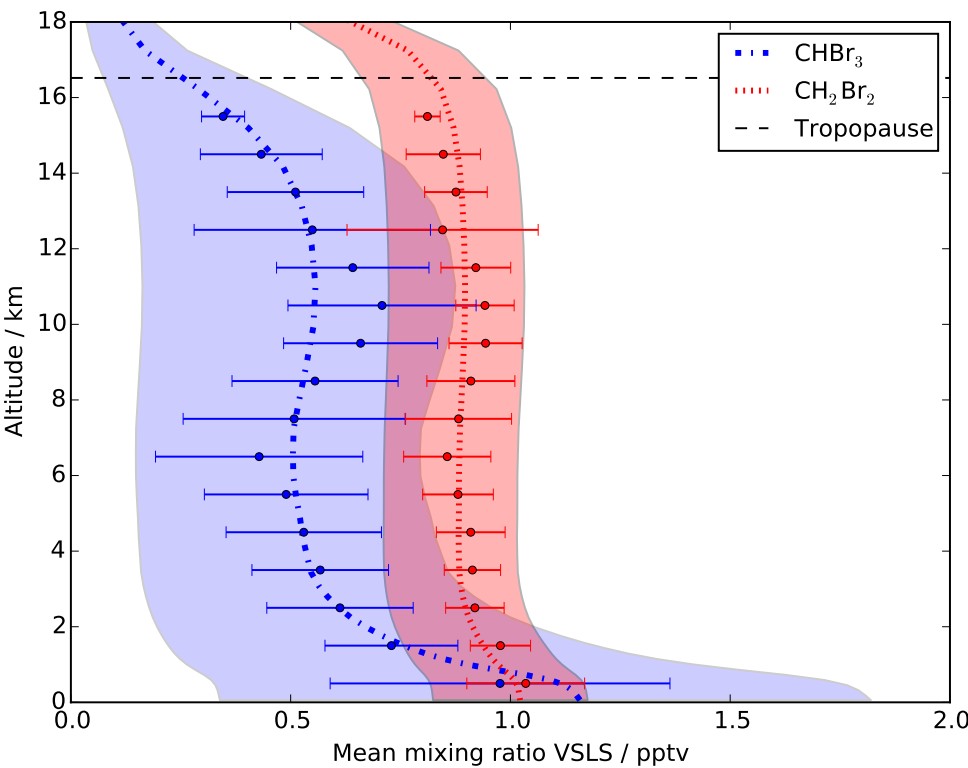

**Figure 12.** Observed (solid circles) and model (dashed-dot line) mean mole fractions of $CHBr_3$ (blue) and $CH_2Br_2$ (red) as a function of altitude, January–February 2014. The horizontal lines associated with each mean observation denotes the range about that mean. The coloured envelopes associated with the model denote the uncertainty based on the bias correction as described in the main text. The black horizontal dashed line denotes the mean model tropopause of 16.5 km.



## Appendix A:  Model Evaluation Using NOAA Surface Mole Fraction Measurements

Figure A.1 is the mean annual percentage bias and associated $r^2$ values betweens modelled and observed $CHBr_3$ and $CH_2Br_2$ at stations in table2. The majority of station sites have a positive model bias with magnitude varying depending on location. Mid-latitude stations (LEF–NWR) have similar bias values of 30–40% (10–20%) for $CHBr_3$ ($CH_2Br_2$). At the tropical sites,

which are comparable with the campaign region, the model bias varies strongly depending on location. KUM and MLO both sit on Hawaii, with KUM and SMO being a near surface coastal station and MLO sitting at an elevated altitude of 3397 m. Model bias calculated for MLO (60%) is much greater than the other two near surface sites (<10%), however it gives the strongest annual correlation with $r^2$ values of 0.75 (0.55) for $CHBr_3$ ($CH_2Br_2$). All coastal sites (with exception of ALT) near emission sources have low $r^2$ values (<0.4) suggesting the model does not capture local variations in emissions well.

Seasonal variations within model bias and correlations of $CHBr_3$ and $CH_2Br_2$ are shown in Table 3. The campaign season of DJF is poorly constrained within the model at all sites with an $r^2$ < 0.5 for both gases. The annual correlation at sites appears to be dominated by other seasons. Within the tropical stations, model bias increases from the annual at KUM to around 20% with no correlation to observed values. MLO and SMO show a similar seasonal bias to the annual indicating the effect to be local to the KUM station site.

Figure A.2 is the observed modelled and seasonal cycle at the tropical station sites (KUM, MLO and SMO) for $CHBr_3$ and $CH_2Br_2$. The model is able to reproduce the seasonal cycle well at all three sites. The emissions at these sites are not scaled seasonally, the phase is representative of the chemistry at these sites. The shorter lived $CHBr_3$ profile is dominates by its loss from photolysis whereas the $CH_2Br_2$ cycle is dominated by oxidation with OH. The amplitude of the seasonal cycle is overestimated in $CHBr_3$ at MLO, and to a lesser extent KUM. This can be indicative of local biases within photolysis loss rates

and/or emissions. The same effect is not shown within the $CH_2Br_2$ suggesting there is not a similar problem associated with OH fields. This is concurrent with a recent multi-decadal analysis of carbon monoxide Mackie et al. (2016) at higher northern latitudes does not support a major problem with similar monthly 3-D fields of OH.





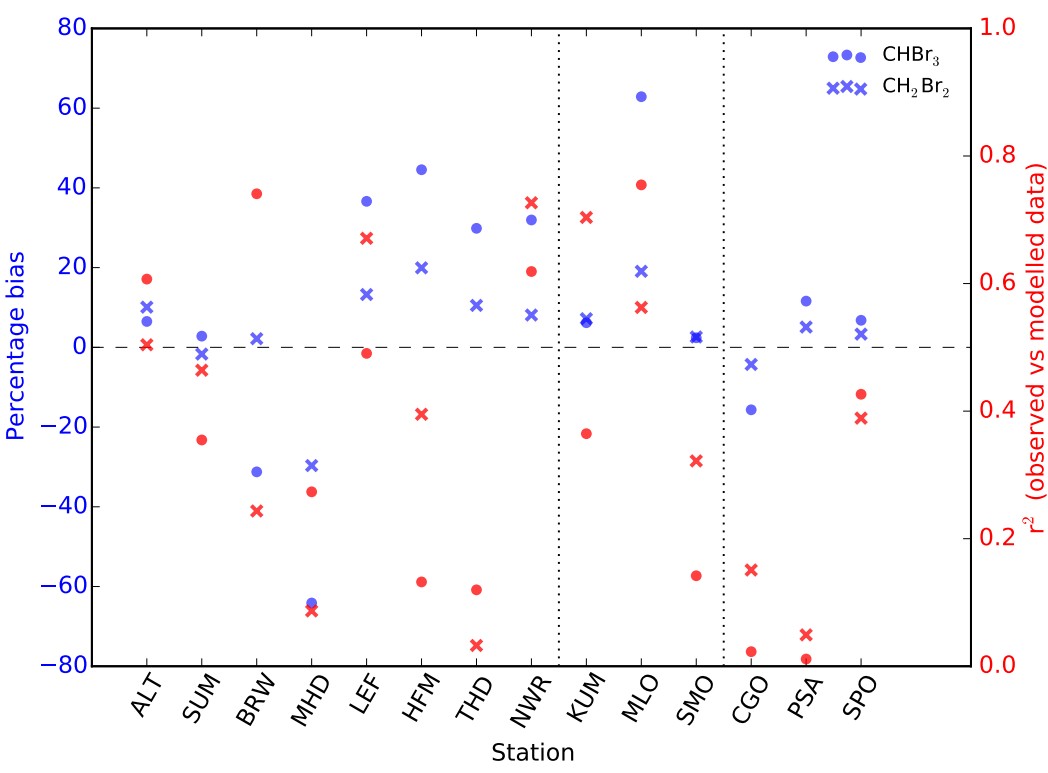

**Figure A.1.** Mean annual percentage model bias (blue) calculated at NOAA ground station sites (Table 2) for $CHBr_3$ (dots) and $CH_2Br_2$ (crosses). The horizontal dashed line denotes zero bias. The right-hand-side y axis describes the ability of the model to reproduce observed variations ($r^2$) (red). The vertical dotted lines define the tropical stations.





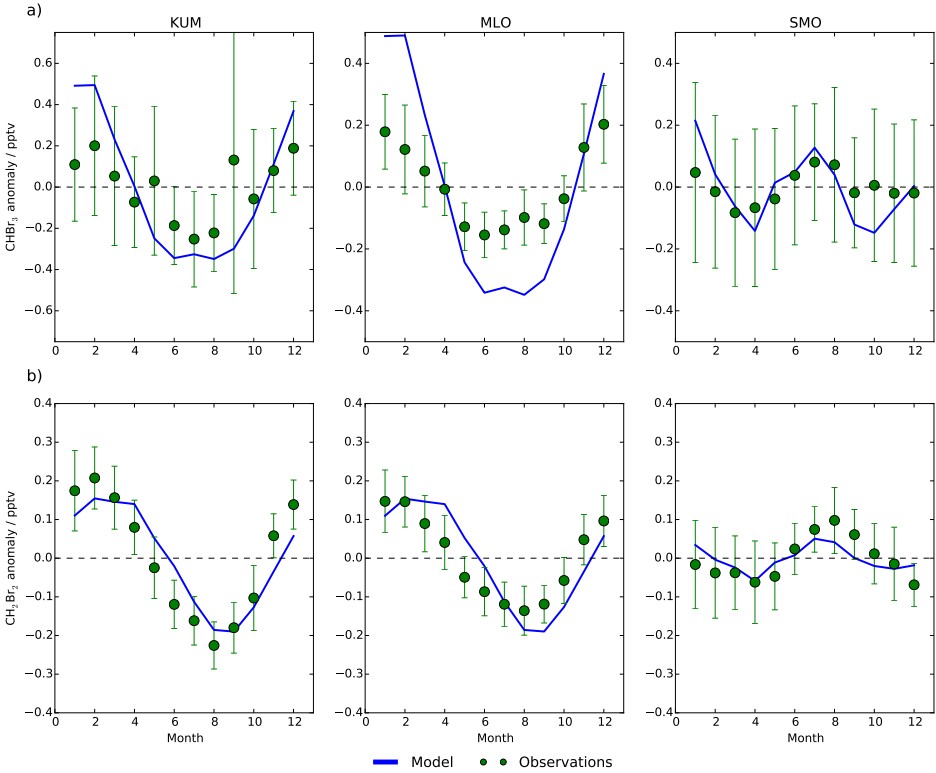

**Figure A.2.** Observed (green) and model (blue) mole fractions of (a) CHBr$_3$ and (b) CH$_2$Br$_2$ at tropical NOAA sites. The seasonal cycle is shown as the climatological monthly mean anomaly calculated by subtracting the annual mean from the climatological monthly mean (pptv). Horizontal bars on observed values denote $\pm 1\sigma$.

## Appendix B:  Evaluation of Model Convection

To evaluate model convection over the marine environment during the CAST and CONTRAST campaigns, we developed a short-lived tagged tracer simulation with an e-folding lifetime comparable to CH$_3$I, as described in section 3.

We emitted CH$_3$I at an equilibrium mole fraction of 1 pptv over ocean regions and applied an atmospheric e-folding lifetime
5 of four days similar to that CH$_3$I in the tropics Carpenter et al. (2014). We can then use the model mole fraction to determine the effective mean age of air parcels throughout the troposphere, and to compare the qualitatively to observed CH$_3$I values collected during the CONTRAST campaign.

Figure B.1 is a comparison of observed CH$_3$I to our CH$_3$I-like tracer. The model can generally reproduce the quantitative vertical distribution of CH$_3$I: a decrease from the surface source up to an altitude of 10–11 km. Above this, there is a 1-2 km
10 altitude region where values are higher than those in the free troposphere, suggestive of outflow from convection. As expected,





the youngest air masses are close to the surface with the ages as young as 5–6 days in the upper troposphere. These ages are indicative of fast convective transport but they are not as young as would be expected from some of the highest observed mole fractions, which are likely due to faster, sub-grid scale, convective transport.

Figure B.2 is the probability distribution of the age of the simulated $CH_3I$-like tracer between 10–15 km altitude. The model captures infrequent fast, large-scale convective transport over the study domain, with ages as young as 3–4 days reaching the upper troposphere. One metric to describe the convective transport is the marine convection index (MCI), following Bell et al. (2002): the ratio of mean upper tropospheric $CH_3I$ (8–12 km) to lower tropospheric $CH_3I$ (0–2.5 km). The CONTRAST observations have an MCI of 0.38 and the corresponding model MCI sampled for these observations is 0.19. The MCI for the model domain for the duration of CONTRAST is 0.28. These values are consistent with those found in Bell et al. (2002) over a similar Pacific regions.

Overall, we find that the model describes the mean convective flow over the region and can capture instances of rapid, large-scale convective transport. Differences in the MCI suggest a significant role for rapid, sub-grid scale vertical transport that are not captured by our coarse model resolution.




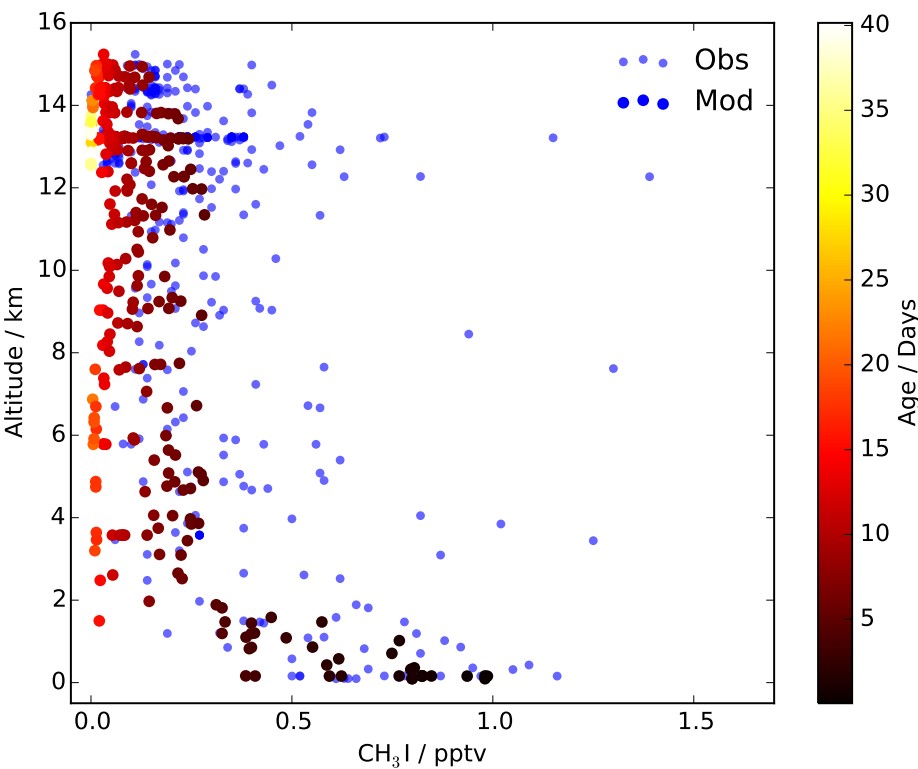

**Figure B.1.** Vertical profiles of observed (blue) and synthetic (coloured as a function of age) $CH_3I$ mole fraction data collected by AWAS during CONTRAIL as a function of altitude.





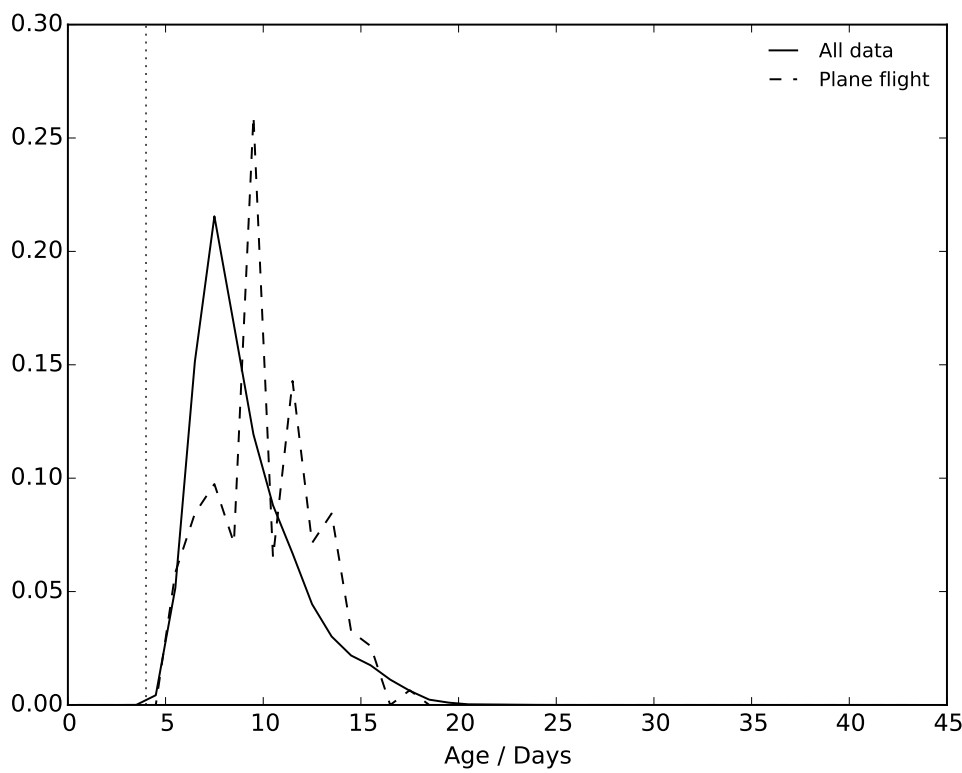

**Figure B.2.** Probability distribution of the physical age of CH$_3$I for the 3-D study domain (solid line) and as sampled by the aircraft (dashed line) between 11–15 km during CONTRAST, 18th January–28th February, 2014. The dotted line indicates 1$\tau_{CH_3I}$ of 4 days.