# Peer review of "Quantifying the vertical transport of CHBr3 and CH2Br2 over the Western Pacific"

_Atmospheric Chemistry and Physics, 2016_

## Referee Comment (RC1) · Anonymous Referee #2 · 28 Dec 2016

General Comment

The paper presents a modelling experiment oriented to quantify and attribute the location of oceanic CHBr3 and CH2Br2 sources reaching the Upper Troposphere (UT) and Tropical Transition Layer (TTL) within the Western Pacific. GEOS-Chem tagged simulations are performed to determine the extent at which coastal and open ocean geographical regions contribute to the CHBr3 and CH2Br2 mixing ratios at different heights, as well as to determine the physical age of air of the air parcels arising from each of the tagged regions. Experiment results are compared to atmospheric measurements performed during CONTRAST and CAST in Jan-Feb 2014, showing a good agreement when the model bias is removed. Overall, I found the study very interesting and of relevance for ACP. It contributes to understand how oceanic sources of brominates substances are transported to the TTL, a transit region with important strato-
spheric injection implications. However, I have some concerns respect to the validity of one of the tagged scenarios and the model validation against NOAA data; and most importantly, I found many of the analysis and discussions given in Sections 4 (Results) and 5 (Discussions and Concluding remarks) very vague and/or requiring stronger evidence that supports them. I suggest being more specific and formal on the explanation of the specific statements highlighted in this review.

Specific Comment I

The emission inventory for CHBr3 and CH2Br2 used in GEOS-Chem is that from Liang et al., 2010, which is based on Warwick et al., 2006. Those emissions scenarios include only oceanic sources (continental emissions are zero), and include a coat-to-ocean enhancement to better fit experimental data. Then, I do not understand why one of your tagged scenarios is for "land", I would have join "land + coastal ocean" into a unique tagged scenario (and compared it to open ocean), as all of the CHBr3 and CH2Br2 prevailing in the land tagged scenario currently belong to emissions from coastal ocean within the Liang et al., 2010 inventory.

P5;L6-8: Please provide a proper reference to the NOAA ETOPO2v2 Global Relief map, and explain how the 2 minute spatial resolution from that database is extrapolated to the 2° x 2.5° horizontal resolution of GEOS-Chem, and how well it compares to the land mask of the model. Also, what is the resolution of the Liang et al., 2010 inventory? "For tracers that spatially overlap we calculate their fractional contribution taking into account the area covered by land or ocean and local emission fluxes". Couldn't it just be done by computing the landfraction of each of the GEOS-Chem grids with coastal emissions?

P5;L16: You give global annual totals of CHBr3 and CH2Br2 emissions for the Liang et al. inventory, but it would be very useful if you could explicitly indicate what fraction of the annual global source is emitted within the modelled region during the Jan-Feb modelled period. Later in the results section there is an explicit reference of the contribution of VSL sources arising from outside the study region, so knowledge of the net emission from the selected region is valuable. Also, here you explicitly mention the imposition of a seasonal cycle, whereas in the Appendix A you state that there is not any seasonal cycle. Please make this point clear. Finally, do you apply any daily profile to the emissions or they are constantly being emitted during the 24 hs of a day?

Fig. 2: The Open Ocean emission includes maximum values above $1.0 \times 10^{-13}$ kg m2 s-1 (correct the units on the figure) which are probably related to the coastal particularities. The rest of the open-ocean is quite constant with a latitudinal dependence as in the Liang et al. emissions inventory. Why did you include those large variable hot-spot into the open-ocean tracers? This certainly increases the open ocean contribution to the overall abundance of $CHBr_3$ and $CH_2Br_2$.

Note that the coastal ocean age of air profile shown in Fig. 9 is very similar to the land profile. Isn't this indicative that GEOS-Chem represents convective transport similarly for the coastal-ocean and land tagged scenarios? The coastal oceans even shows more aged air-masses that the land? Could you explain this? Could you also explain by how much does the coastal ocean age of air contribute to the whole ocean (open+coastal) profiles?

Specific Comment II

There are some details on the NOAA VSLS validation that should be explicitly stated in the text. From Table 2 and Appendix A, it becomes evident that none of the 14 NOAA stations is located in the Western Pacific, the area of study. Indeed, the pacific stations are located either in Hawaii, Australia or Samoa Island, well outside the study region. P6;L17: "The model generally has less skill at reproducing observations collected at coastal sites close to emission sources". Then, this constitutes an additional factor which must be considered when computing the uncertainties of the "coastal ocean" contribution to the overall $CHBr_3$ and $CH_2Br_2$ abundances in the MBL, FT and TTL. This is not explicitly mentioned in the text.

P1;L7: "The model has a mean positive bias of 30% that is larger near the surface reflecting errors in the poorly constrained prior emission estimates". P6;L23 "In general, GEOS-Chem has poorer skill at reproducing observed near-surface variations, reflecting errors in prior emissions". There must be other factors affecting the model results: if only a bias on VSL sources exist, then the bias should remain constant in height. How do you relate these statements with the fact that the CONTRAST and CAST campaigns occur within a region of large coastal areas, so results for coastal ocean tags might not be so reliable.

P30;L4: "At the tropical sites, which are comparable with the campaign region, the model bias varies strongly depending on location". This could be indicative of the large variability of convective events within the tropical sites, besides the mentioned errors on prior emissions.

Specific Comment III

The vertical profile of modelled and measured CHBr3 and CH2Br2 abundances is not given until the very last figure (Fig. 12). Many panels showing the vertical variation of the model bias, as well as the percentage contribution of each of the tagged scenarios, are shown before the absolute vertical profile is given. I imagine Fig. 12 is shown at the very end of the paper because the authors preferred to present it after all the analysis of sources, uncertainties and associated bias has been described, buy It would be very useful to have it placed early in the text, so the reader has an absolute value in mind when all differences, bias and percentages are computed. Additionally, many of the initial comments, such as the "S" shape profile for CHBr3, could be visualized at first glance.

Fig. 12: Have you thought on showing as a separate panel the original results without the corrected-bias procedure (and perhaps showing the model standard deviation within the WP region). This would help to visualize how well does the model in reproducing the observed values of CAST/CONTRAST or any other campaign. The Bias

correction is helpful to improve the estimation of the VSL burden in the TTL and it impacts on stratospheric injection, but the procedure is still dependent on the model capability on reproducing the measured data.

Further Comments:

P1;L17 (abstract): "and a mean (range) Bry mole fraction of 3.14 (1.81–4.18) pptv to the upper troposphere". This sentence in the abstract gives the impression that you have quantified both Source Gas and Product Gas bromine, whereas you have only presented results for carbon-bonded source gases. This confusion is only clarified when reading the conclusions. Please rephrase in the abstract to make it clear.

I was surprised the MS does not give any single mention of the contribution of minor VSLS (such as CH2BrCl, CHBr2Cl, etc.) to the atmosphere. Even when minor VSLS are not included in the model experiments and no experimental data is presented, at least a mention of their relevance should be given in the MS.

The description of the Age of Air (A) computation is quite confusing. What magnitude of the surface boundary condition increases linearly with time? Is it the area B, or the vmr of the tracer within the transported air-mass? Also, Is there any physical interpretation for the scaling factor and its value? Note that a fraction of the final sentences of the paragraph describing the CH3I tracer belongs to the Results section. Please also briefly explain how the Convective Mass Flux (CMF) is computed in the model.

P5;L27: you mentioned that the usage of age of air is useful "in the absence of reliable bottom-up emissions inventories". In my opinion, the use of age of air simulations helps to understand the rapid convection independently of the existence (or not) of bottom-up inventories. Please note that Ziska et al., (2013) presented a bottom-up inventory for VSL species.

P6;L15: How do you compute the 30-60% value of the seasonal variation?

Fig. 4 shows there is a larger bias for CHBr3 at the surface, but for CH2Br2 the bias

is larger at higher heights. This is not explained in detail. Also, why Figure 4 x-axis title indicates "tagged model vs. observed VSL". Isn't this comparison considering all oceanic (coastal + open) plus land sources altogether? If any specific tagged region is considered, that should be explained in the text.

P7;L13-L16: "Averaged over the campaign, coastal and terrestrial sources of CHBr3 show little influence above 6 km". Then, what is controlling CHBr3 abundance over 6 km. Only Open ocean? Also, "At the TTL, averaged over the campaign study, CH2Br2 mole fractions range 0.1–0.3 ppt mainly due to smaller magnitude of ocean emissions compared to CHBr3. Coastal and terrestrial sources contribute up to 0.1 ppt of CH2Br2 in the TTL". And the remaining CH2Br2 in the TTL, where does it comes from?

P7;L30: "The longer lifetime of CH2Br2 mean that these mole fractions have a greater influence over the campaign profile compared to CHBr3." I found this statement very vague or unspecific. What do you mean by "greater influence over the campaign profile"? Do you mean the profile does not decay so rapidly? The campaign profile for each species certainly depends on the lifetime, but I do not understand how the "influence" of the lifetimes from one species to the other can be quantified.

P8;L6: "The oldest ages, which approach the time of the study period, reflect the accumulation of near-zero mole fractions." I do not understand the meaning nor the implications of this sentence.

P8;L19: "Despite intensive measurements around coastal land masses of the region, CAST did not very well capture coastal emissions." Couldn't it be possible that GEOS-Chem did not represent properly the age of air for this coastal areas?

P8;L24: "The only exception is at the near-surface where land emissions dominate the older age profile." Wouldn't coastal emissions also be contributing to the aged profile near the surface?

P9;L9-L10: "Based on average observed surface values of CHBr3 (1.13 ppt) and

CH2Br2 (1.02 ppt) over the campaign we infer that 40% and 86% of these emitted gases, respectively, are directly injected into the TTL over our study domain". I completely disagree with this statement and found it inconsistent to what is being described above. As we move upward in the troposphere, a larger fraction of the VSLS abundance cannot be explained without considering the contribution from source regions "outside" the study domain. Thus, is expected that from the 0.46/1.13 and 0.88/1.02 ratios of TTL/Surface vmr, there is a contribution in the numerator arising from other sources outside the domain . . . thus less than that percentage is directly being transported to the TTL within the study domain.

P9;L9-13: Fernandez et al., 2014, also performed different sensitivity studies including only CHBr3, only CH2BR2 and other minor VSLS in a CTM, and determined the amount of CHBr3 and CH2Br2 being decomposed before reaching the TTL within the global tropics and the WP.

P10;L5: "Tropospheric measurements of CH2Br2,. . ., are dominated by sources from before the campaign". P7;L25: "The remaining contributions are representative of emissions before the campaign period". How do you attribute those values to emissions before the campaign period?. Although it is expected that the species with longer lifetime will have a longer-lasting contribution until its final decay, the statement should be based on any of the results presented in the text.

P10L16: "Our flux estimate for CHBr3 is lower than previous studies that have reported values closer to 50%." 50% of what? Of the overall inorganic bromine burden or respect to the Surface CHBr3 abundance.

Technical Comments

P1;L6: "32%–37% of CHBr3 observed variability and 15%-45% of CH2Br2 observed variability". Rephrase to use only observed variability only once.

P2;L23, and elsewhere in the MS: Whenever many references are being cited, they

should be ordered chronologically.

P3;L21: "other non-WAS halocarbon data, not analyzed here, have only recently become available". If you include this type of statement, then you should properly reference it.

P3;L29, "between the two WAS systems and two accompanying GC/MS instruments" it is not clear if the percentage error applies to CHBr3 and CH2Br2 species or to the WAS and GC/MS instruments.

P4;L2: You should indicate what the GMAO-FP Office is for the people not familiar with meteorological data, and properly reference it.

P4;L9, correct the subtitle with "and" between species.

P4;L21: What is the top model pressure and or height?

P5;L1: "...the following temperature (T) ...by OH (Sander et al., 2011)

P6;L19: The Equation used for computing the bias uses model_i and mod_i as variables, which I do understand belongs to the same value

P7;L14: by virtue of its longer atmospheric lifetime.

P7;L26: "Figure 8 shows the same as Figure 7 but for CH2Br2." Please, rephrase.

P8;L9: "We using our CH3I-like tracer that air masses can be transported to the TTL within 3–5 days but these are infrequent events (Appendix B)." rephrase.

P8;L28: replace 20-40 to 24-48

P9;L7: They estimated...

P9;L14: but does not take

Appendix B. Indicate if Figure B.1 includes data sampled only at the same times and locations as the CONTRAST (not CONTRAIL) flight tracks, or if the whole study domain

has been considered.

Fig. 1 and Fig 3: M (Manus) is not included in these figures but it is on the others.

Fig.2: replace "in to" by "into"

Fig. 9: Indicate whether the values were averaged within the whole study domain or only the flight tracks.

Fig.10: What is the "percentage of occurrence rate"? rate of what?
* * *

---

## Referee Comment (RC2) · Anonymous Referee #3 · 6 Jan 2017

The manuscript investigates the vertical transport of very short-lived halocarbons over the Western Pacific based on model simulations and aircraft measurements. Bromoform and dibromomethane observations from two aircraft campaigns are linked to simulations of tagged tracers and age of air. The study is in general of interest to the readership of ACP. However, the analysis is not presented clearly and major uncertainties and assumptions are not discussed appropriately. Moreover, the discussion of the results is confusing in many places. I suggest publication after major revisions addressing the comments listed below.

1) Results in section 4.2 regarding the tagged-VSLS model output depend very strongly on the chosen emission scenario. Most of the information presented here (i.e., the amount of coastal versus open ocean emissions contributing to upper air mixing ratios) could be quite different for another emission scenario. This aspect is not addressed or

[Figure]

discussed at all in the manuscript. Given the large differences between the different emission scenarios and existing research investigating those differences and the implications for atmospheric mixing ratios (Hossaini et al., 2013; Hossaini et al., 2016) a proper discussion is required. Ideally, the study should be carried out based on at least one more emissions scenario in order to understand the uncertainties resulting from the assumptions made here.

2) The choice of the emission scenario is not discussed. Why top-down and not bottom-up? Which scenario is thought to be the most realistic in this region? Why is this scenario used if the simulated surface mixing ratios show large deviations to the observations? Could these deviations be minimized for a different (lower) emission scenario?

3) What would cause land sources of CHBr3 and CH2Br2? In the introduction, only marine sources are discussed, but later the reader is confronted with the land tagged tracer and its contribution to the observed mixing ratio.

4) The discussion of the model evaluation (section 4.1) needs to be improved. How large are the relative deviations between model and observations. If the bias is mostly a result of the emissions used, than the relative differences should stay constant with height. If however, the relative differences increase or decrease with height this would indicate errors introduced by the transport scheme of the model. Even tough, there are six panels used to discuss the comparison such conclusions are currently not possible.

5) Please provide the model resolution. At the moment only the resolution of the meteorological input data is given. Is this the same as the model resolution and the resolution of the output data? How would this quite coarse resolution (2° x 2.5°) impact the results? In particular, how would this impact the model-based analysis of the observations?

6) Please improve description and discussion of Figures 9, 10 and 11. It is difficult to understand what has been done and why some of the statements are made. See also

detailed comments further below.

Minor comments

Page 5, line 4-12. Please explain Figure 2. Are the tagged tracer regions shown or are the tracer regions combined with the emission scenario shown? How do you end up with 20 tagged tracers? Seven for $CHBr_3$ and seven for $CH_2Br_2$ and the rest for total and background?

Page 5, line 15. Please explain what 'de-seasonalized monthly means'? Are you using annual means? Or interannual anomalies plus mean values?

Page 5, line 26. This sentence makes no sense. You use age of air simulations because you have no reliable emission inventory? But then the other half of the analysis is based on one emission inventory? Furthermore, should this sentence suggest that only the bottom-up inventories are unreliable while the top-down are not?

Page 6, line 21-22. Please explain how the amount of explained variability is estimated.

Page 7, line 17. How were those percentage contributions calculated? Transform numbers from Figure 5 into relative numbers and then apply them to the observations? Here and at other places, the methodology is not clear and the reader has to guess what exactly has been done.

Page 8, line 5. I don't understand how the discussion of Figure 9 (which shows age of air as a function of source region but no emissions or mixing ratios) allows such a statement. Or is here information from other earlier analysis used? Same for line 8.

Page 8, line 12. The text says that 53% of what reaches the TTL comes from the open ocean? From other parts of the manuscript, I had the impression that the large majority comes from the open ocean? Please clarify.

Figure 9: Comparing the lines for ocean, open ocean and coastal ocean, I wonder if the coastal and open ocean together should give the ocean age of air? However, the

total ocean (blue line) shows the youngest age of all. Please clarify.

---

## Referee Comment (RC3) · Q. Liang (Referee) · 12 Jan 2017

Butler et al. (2016) presented a modeling analysis of the transport of very-short-lived bromocarbons from the surface to the UT/LS over the Western Pacific. This analysis is based on the GEOS-Chem simulation of CHBr3, CH2Br2, age of air tracer and aircraft measurements from the CAST and CONTRAST campaigns. After reading through the manuscript, I have to say that I share similar concerns with the two other reviewers. Here, I am not going to repeat many of the issues raised by the other two reviewed, but just stating the major issues with the current model design and approach:

1. The use of Liang et al. (2010) emissions. I don't understand why there were emissions over the land, as the Liang et al. emissions scheme only specifies emissions from Open Ocean and coastal regions. While the original inventory was derived on 2x2.5

horizontal resolution, I provided a refined emissions inventory on 1x1 degree resolution to the GEOS-Chem group. Could it be possible when the 1x1 degree emissions were regrided to 2x2.5, emissions appeared to occur over the island landmasses as a result of coarse resolution? Whatever the reason was, the use of land tagged emissions tracers for CHBr3 and CH2Br2 and the reference of terrestrial sources of these gases throughout the manuscript, in my view, are not accurate and lead to wrong impression that land could be a source of these oceanic-originated compounds. Second, the Liang et al. (2010) emissions inventory was originally derived for stratospheric bromine budget purposes (therefore without much attention to fine-tuning the surface emissions details, e.g. longitudinally invariable and simple treatment of Open Ocean vs. Coasts), with no observations over the western Pacific to constrain surface emissions in that region. As shown by Hossaini et al. (2013), the Ziska bottom-up inventory is a much more skillful and a more appropriate choice of emissions for the Western Pacific region. Quantifying the relative importance of open ocean emissions vs. coastal sources using the Liang et al. (2010) emissions scheme for the Western Pacific region, which is one of the main focus of this paper, does not provide a credible estimate.

2. Page 8, Line 29 – Page 9, Line 9. I have to say I don't see the meaning of the use of modeled profiles of CHBr3 and CH2Br2 by applying a vertical uniform correction of model biases to quantify SGI and PGI.

i) First, the estimated injection of PGI based on model corrected profiles is not correct. Why use the model? The model, even after correction, still shows low biases for CHBr3 and CH2Br2 at 10-12 km. In fact, shouldn't the observation-based organic Br be the true PGI value?

ii) While it was not explained in the text, my guess is that the authors use the difference of Br value at the surface and that at the TTL to calculate PGI. This is not a correct approach in my view. As show in Liang et al. (2010) , a significant fraction of the inorganic Br produced from CHBr3 and CH2Br2 degradation are removed by large-scale precipitation in the lower troposphere and never makes to the UT.

3. Same as the other reviewers, I also find the use of idealized age of air, in particular the results presented in Figure 11, hard to interpret.

---

## Author Comment (AC1) · 15 Aug 2017

**Quantifying the vertical transport of CHBr$_3$ and CH$_2$Br$_2$ over the Western Pacific**

Robyn Butler et al

We thank the two anonymous reviewers and to Q. Liang for their constructive comments. These have helped to improve the revised manuscript. Below are our responses to each individual reviewer comment denoted in italics. We apologise for the delay in responses. Some of the comments needed to be addressed with additional model runs.

I have attached 3 additional Figures in response to comments, and an updated version of the manuscript as a supplement.

**Review report 1**

**Specific Comment I**

*The emission inventory for CHBr3 and CH2Br2 used in GEOS-Chem is that from Liang et al., 2010, which is based on Warwick et al., 2006. Those emissions scenarios include only oceanic sources (continental emissions are zero), and include a coat-to ocean enhancement to better fit experimental data. Then, I do not understand why one of your tagged scenarios is for "land", I would have join "land + coastal ocean" into a unique tagged scenario (and compared it to open ocean), as all of the CHBr3 and CH2Br2 prevailing in the land tagged scenario currently belong to emissions from coastal ocean within the Liang et al., 2010 inventory.*

**RESPONSE COMMON TO REVIEWER 1 and 2**

The land tracer only appeared because of the mask we used to define coastal and open ocean sources. Land tracers have been included in the coastal ocean tracer. We have now changed the manuscript so the land tracer does not feature in the discussion.

*P5;L6-8: Please provide a proper reference to the NOAA ETOPO2v2 Global Relief map, and explain how the 2 minute spatial resolution from that database is extrapolated to the 2x2.5 horizontal resolution of GEOS-Chem, and how well it compares to the land mask of the model. Also, what is the resolution of the Liang et al., 2010 inventory? "For tracers that spatially overlap we calculate their fractional contribution taking into account the area covered by land or ocean and local emission fluxes". Couldn't it just be done by computing the land fraction of each of the GEOS-Chem grids with coastal emissions?*

We have now added a reference for these data, and made the associated changes to the manuscript. We have also added and changed the text to improve the description of the tracer calculations: "Fractional contributions (R) of tracers are calculated by assigning each 2-minute cell (c) in a model grid box a value of 1 depending on tracer

definitions, 0 if it doesn't. These cells are then calculated as a fraction of the total number in each model grid box. $R = \Sigma N_{c=1}/N_{ctotal}$."

We developed a land mask from the bathymetry data because the coarser GEOS-Chem land mask does not include the local geography where the campaigns were focused (Guam, Palau and Chuuk).

*P5;L16: You give global annual totals of CHBr3 and CH2Br2 emissions for the Liang et al. inventory, but it would be very useful if you could explicitly indicate what fraction of the annual global source is emitted within the modelled region during the Jan-Feb modelled period. Later in the results section there is an explicit reference of the contribution of VSL sources arising from outside the study region, so knowledge of the net emission from the selected region is valuable. Also, here you explicitly mention the imposition of a seasonal cycle, whereas in the Appendix A you state that there is not any seasonal cycle. Please make this point clear. Finally, do you apply any daily profile to the emissions or they are constantly being emitted during the 24 hs of a day?*

We have added regional seasonal totals for the study period.

We thank the reviewer for spotting the inconsistency in the text about the seasonal cycle. The full chemistry GEOS-Chem has an imposed seasonal cycle, but we do not use that information in our tagged simulation. There is no diurnal cycle to the emissions, they remain constant for 24hs. We have modified the text accordingly to clarify this.

*Fig. 2: The Open Ocean emission includes maximum values above 1.0x10-13 kg m2 s-1 (correct the units on the figure) which are probably related to the coastal particularities. The rest of the open-ocean is quite constant with a latitudinal dependence as in the Liang et al. emissions inventory. Why did you include those large variable hot-spot into the open-ocean tracers? This certainly increases the open ocean contribution to the overall abundance of CHBr3 and CH2Br2.*

First, we thank the reviewer for unit correction.

We included the elevated value because they fell within the open oceanic tracer mask. To address the reviewer concern, we completed a model run that assigned these elevated value into the coastal ocean tracer.

The revised Figure (shown in Figure 1) below shows elevated coastal contributions over coastal regions in the vertical profile but there remains a dominant contribution from the open ocean emissions for both gases. This revised calculation changes some parts of the discussion of results but the general findings remain the same, e.g. coastal emissions play a larger role at higher altitudes. We have modified the text to reflect these revised calculations.

*Note that the coastal ocean age of air profile shown in Fig. 9 is very similar to the land profile. Isn't this indicative that GEOS-Chem represents convective transport similarly for the coastal-ocean and land tagged scenarios? The coastal oceans even shows more aged air-masses that the land? Could you explain this? Could you also explain by how much does the coastal ocean age of air contribute to the whole ocean (open+coastal) profiles?*

Following this reviewers recommendation (as described above) we have removed the land tracer from the discussion.

**Specific Comment II**

*There are some details on the NOAA VSLS validation that should be explicitly stated in the text. From Table 2 and Appendix A, it becomes evident that none of the 14 NOAA stations is located in the Western Pacific, the area of study. Indeed, the pacific stations are located either in Hawaii, Australia or Samoa Island, well outside the study region. P6;L17: "The model generally has less skill at reproducing observations collected at coastal sites close to emission sources". Then, this constitutes an additional factor which must be considered when computing the uncertainties of the "coastal ocean"*

*contribution to the overall CHBr3 and CH2Br2 abundances in the MBL, FT and TTL. This is not explicitly mentioned in the text.*

This reviewer's comments are well taken. The purpose of evaluating the model with the NOAA VSLS data was to quantify and report model performance and not intended to provide additional data over the Western Pacific. We will make that explicit in the text. We will also relate our findings about the coastal NOAA data to our analysis of the aircraft data over the western Pacific.

*P1;L7: "The model has a mean positive bias of 30% that is larger near the surface reflecting errors in the poorly constrained prior emission estimates". P6;L23 "In general, GEOS-Chem has poorer skill at reproducing observed near-surface variations, reflecting errors in prior emissions". There must be other factors affecting the model results: if only a bias on VSL sources exist, then the bias should remain constant in height. How do you relate these statements with the fact that the CONTRAST and CAST campaigns occur within a region of large coastal areas, so results for coastal ocean tags might not be so reliable.*

**RESPONSE COMMON TO REVIEWER 1 and 2**

In Figure 4 (Model Evaluation) we now include a panel describing the relative model error. We find this model error is consistent with altitude for both gases relating to a constant offset in model values. We explicitly mention this result in the manuscript text. We also acknowledge that values outside of the mean value could be indicative of model transport error.

*P30;L4: "At the tropical sites, which are comparable with the campaign region, the model bias varies strongly depending on location". This could be indicative of the large variability of convective events within the tropical sites, besides the mentioned errors on prior emissions*

The reviewer raises a good point. We now include a statement about this point where

we discuss the NOAA station data.

**Specific Comment III**

*The vertical profile of modelled and measured CHBr3 and CH2Br2 abundances is not given until the very last figure (Fig. 12). Many panels showing the vertical variation of the model bias, as well as the percentage contribution of each of the tagged scenarios, are shown before the absolute vertical profile is given. I imagine Fig. 12 is shown at the very end of the paper because the authors preferred to present it after all the analysis of sources, uncertainties and associated bias has been described, buy It would be very useful to have it placed early in the text, so the reader has an absolute value in mind when all differences, bias and percentages are computed. Additionally, many of the initial comments, such as the "S" shape profile for CHBr3, could be visualized at first glance.*

Good point. We have modified Figure 4 so that the top panel shows the vertical distribution of model and observations from campaigns in the top panel, the middle panel show observations vs. model scatter, and the bottom panel shows relative model error over the vertical profile.

*Fig. 12: Have you thought on showing as a separate panel the original results without the corrected-bias procedure (and perhaps showing the model standard deviation within the WP region). This would help to visualize how well does the model in reproducing the observed values of CAST/CONTRAST or any other campaign. The Bias correction is helpful to improve the estimation of the VSL burden in the TTL and it impacts on stratospheric injection, but the procedure is still dependent on the model capability on reproducing the measured data.*

We believe our revised Figure 4 now addresses this comment.

**Further Comments**

*P1;L17 (abstract): "and a mean (range) Bry mole fraction of 3.14 (1.81–4.18) pptv to*

*the upper troposphere". This sentence in the abstract gives the impression that you have quantified both Source Gas and Product Gas bromine, whereas you have only presented results for carbon-bonded source gases. This confusion is only clarified when reading the conclusions. Please rephrase in the abstract to make it clear.*

Agreed. We have changed the abstract to read "and a mean (range) Bry mole fraction of 3.14 (1.81–4.18) pptv from source gases to the upper troposphere".

*I was surprised the MS does not give any single mention of the contribution of minor VSLS (such as CH2BrCl, CHBr2Cl, etc.) to the atmosphere. Even when minor VSLS are not included in the model experiments and no experimental data is presented, at least a mention of their relevance should be given in the MS.*

Agreed. We have now mentioned additional VSLS in the introduction of the MS. "VSLS include gases such as bromoform (CHBr3), dibromomethane (CH2Br2), bromochloromethane (CH2BrCl), dibromochloromethane (CHBr2Cl) and bromodichloromethane (CHBrCl2). Here, we focus on CHBr3 and CH2Br2...".

*The description of the Age of Air (A) computation is quite confusing. What magnitude of the surface boundary condition increases linearly with time? Is it the area B, or the vmr of the tracer within the transported air-mass? Also, Is there any physical interpretation for the scaling factor and its value? Note that a fraction of the final sentences of the paragraph describing the CH3I tracer belongs to the Results section. Please also briefly explain how the Convective Mass Flux (CMF) is computed in the model.*

This comment is comment to both reviewers so we have obviously not described it clearly enough. Here, we address the reviewer questions and rewrite the confusing text in the manuscript.

The surface boundary condition, $B$, is effectively a surface VMR of each tracer at its emission source that linearly increases with time. An air mass initially has a VMR that reflect the time it was in contact with $B$, but get progressively smaller as time

progresses due to atmospheric mixing. The scaling factor $f$ is a value that helps relate the increasing tie to the corresponding change in VMR.

CMF is from the meteorological fields. We have included a statement in the description of meteorological fields stating that this is where the CMF and dynamic tropopause height is from.

*P5;L27: you mentioned that the usage of age of air is useful "in the absence of reliable bottom-up emissions inventories". In my opinion, the use of age of air simulations helps to understand the rapid convection independently of the existence (or not) of bottom-up inventories. Please note that Ziska et al., (2013) presented a bottom-up inventory for VSL species.*

Agree. In the absence of reliable emission inventories, the age of air simulation allows us to study the role of atmospheric transport on the distribution of trace gases. We have clarified that point in the manuscript. "We use the age of air simulation to understand the role and frequency of rapid convective systems to transport short-lived halogenated compounds to the TTL, independent of emission inventories."

*P6;L15: How do you compute the 30-60% value of the seasonal variation?*

This reflect the Table of squared Pearson correlation coefficients.

*Fig. 4 shows there is a larger bias for CHBr3 at the surface, but for CH2Br2 the bias is larger at higher heights. This is not explained in detail. Also, why Figure 4 x-axis title indicates "tagged model vs. observed VSL". Isn't this comparison considering all oceanic (coastal + open) plus land sources altogether? If any specific tagged region is considered, that should be explained in the text.*

Following this reviewer's comment we have modified Figure 4 (as described above) to show the relative model error to better quantify model error. Our results show that the median relative model error for both gases does not significantly change with altitude. Variations around these median values suggest model transport errors. We

have changed our discussion of model evaluation highlighting a role for atmospheric transport error.

Following the reviewer suggestion, we have changed the x-axis title to say 'Total tagged VSLS / pptv' to indicate that all tracers are included.

*P7;L13-L16: "Averaged over the campaign, coastal and terrestrial sources of CHBr3 show little influence above 6 km". Then, what is controlling CHBr3 abundance over 6 km. Only Open ocean? Also, "At the TTL, averaged over the campaign study, CH2Br2 mole fractions range 0.1–0.3 ppt mainly due to smaller magnitude of ocean emissions compared to CHBr3. Coastal and terrestrial sources contribute up to 0.1 ppt of CH2Br2 in the TTL". And the remaining CH2Br2 in the TTL, where does it comes from?*

Due to the changes in emissions for the tagged tracers following this reviewer's comment, as described above, we have already revised the discussion to address this comment. The missing CH2Br2 is from the open ocean. We have revised the manuscript to clarify this point. "Coastal sources contribute up to 0.1 ppt of $CH_2Br_2$ in the TTL with the remaining originating from an open ocean source."

*P7;L30: "The longer lifetime of CH2Br2 mean that these mole fractions have a greater influence over the campaign profile compared to CHBr3." I found this statement very vague or unspecific. What do you mean by "greater influence over the campaign profile"? Do you mean the profile does not decay so rapidly? The campaign profile for each species certainly depends on the lifetime, but I do not understand how the "influence" of the lifetimes from one species to the other can be quantified.*

We agree that this sentence is vague. We are referring to the influence of emissions prior to the campaign period. The tagged model only represents emissions from the Jan/Feb period of the research campaign, and the tagged CH2Br2 indicates that CAST and CONTRAST measurements are dominated by emissions before Jan/Feb 2014. The 'influence' refers to pre-campaign emissions on the CH2Br2 and CHBr3 campaign profile. Compared with CHBr3, CH2Br2 is influenced by emissions pre-campaign,

whereas CHBr3 is mainly influenced by emissions during the campaign due to their different decay profiles. We have modified the manuscript to clarify this point: "The longer lifetime of CH2Br2 results in a greater influence of emissions prior to the campaign period. In contrast, atmospheric CHBr3 is dominated by emissions during the campaign period."

*P8;L6: "The oldest ages, which approach the time of the study period, reflect the accumulation of near-zero mole fractions." I do not understand the meaning nor the implications of this sentence.*

We hope that some of this confusion will be resolved with a better description of the age of air calculation that both reviewers highlighted.

This sentence refers to the VMR of the age calculation.

Referring to the text above, an air mass initially has a VMR that reflect the time it was in contact with the ocean boundary B, but get progressively smaller as time progresses due to atmospheric mixing. Accumulation of near-zero VMRs refers to the oldest air parcels. The statement explains that the older age profile in comparison to the age calculation. In any case, on reflection we felt it did not add to the discussion so we have removed it.

*P8;L19: "Despite intensive measurements around coastal land masses of the region, CAST did not very well capture coastal emissions." Couldn't it be possible that GEOSChem did not represent properly the age of air for this coastal areas?*

Agreed. This is potentially a resolution issue that we have now acknowledged in the discussion.

*P8;L24: "The only exception is at the near-surface where land emissions dominate the older age profile." Wouldn't coastal emissions also be contributing to the aged profile near the surface?*

This effect has become obsolete with the modified tracers and age of air run. At higher

altitudes, this effect is seen and linked to coastal ocean emissions.

*P9;L9-L10: "Based on average observed surface values of CHBr3 (1.13 ppt) and CH2Br2 (1.02 ppt) over the campaign we infer that 40% and 86% of these emitted gases, respectively, are directly injected into the TTL over our study domain". I completely disagree with this statement and found it inconsistent to what is being described above. As we move upward in the troposphere, a larger fraction of the VSLS abundance cannot be explained without considering the contribution from source regions "outside" the study domain. Thus, is expected that from the 0.46/1.13 and 0.88/1.02 ratios of TTL/Surface vmr, there is a contribution in the numerator arising from other sources outside the domain . . . thus less than that percentage is directly being transported to the TTL within the study domain.*

Agreed. We simplified the issue assuming that values in the TTL are directly representative of the values at the surface. This is not a valuable assumption. We have removed this part of the discussion.

*P9;L9-13: Fernandez et al., 2014, also performed different sensitivity studies including only CHBr3, only CH2BR2 and other minor VSLS in a CTM, and determined the amount of CHBr3 and CH2Br2 being decomposed before reaching the TTL within the global tropics and the WP.*

We refer to this paper in the introduction, but have specifically referenced Navarro et al., 2015, as it is an additional study using this campaign data.

*P10;L5: "Tropospheric measurements of CH2Br2,. . ., are dominated by sources from before the campaign". P7;L25: "The remaining contributions are representative of emissions before the campaign period". How do you attribute those values to emissions before the campaign period?. Although it is expected that the species with longer lifetime will have a longer-lasting contribution until its final decay, the statement should be based on any of the results presented in the text.*

This is relative to the experimental method. Tagged emissions are only tagged from 01/01/2014 therefore the remaining percentage difference between the tagged VSLS tracers and the total VSLS tracer. The total tracer is based on a 12-month spin up from the emission scenario with the 'total ocean' emissions included. The VSLS VMR that is not represented by the tagged tracers will be attributed to 'background' VMR from the spin up file, which represents the VMR from before the campaign. We have clarified this in the method and discussion sections.

*P10L16: "Our flux estimate for CHBr3 is lower than previous studies that have reported values closer to 50%." 50% of what? Of the overall inorganic bromine burden or respect to the Surface CHBr3 abundance.*

Now we have excluded the discussion about SGI from source gases this comment has been deleted from the discussion.

**Technical Comments**

We agree with all the technical comments and have made the recommended changes.

**Review report 2**

*1) Results in section 4.2 regarding the tagged-VSLS model output depend very strongly on the chosen emission scenario. Most of the information presented here (i.e., the amount of coastal versus open ocean emissions contributing to upper air mixing ratios) could be quite different for another emission scenario. This aspect is not addressed or discussed at all in the manuscript. Given the large differences between the different emission scenarios and existing research investigating those differences and the implications for atmospheric mixing ratios (Hossaini et al., 2013; Hossaini et al., 2016) a proper discussion is required. Ideally, the study should be carried out based on at least one more emissions scenario in order to understand the uncertainties resulting from the assumptions made here.*

Agreed. Instead of using different emission scenarios (none of which convincingly reproduce observations) we decided to focus on using the age of air calculation to investigate the influence of ocean emission regions independent of the emission scenario. However, have revised the discussion section to acknowledge different emission scenarios.

*2) The choice of the emission scenario is not discussed. Why top-down and not bottom-up? Which scenario is thought to be the most realistic in this region? Why is this scenario used if the simulated surface mixing ratios show large deviations to the observations? Could these deviations be minimized for a different (lower) emission scenario?*

**FIGURE COMMON TO REVIEWER 2 and 3**

Our preliminary work used the alternative Ziska emission scenario. Here we have reported results from the associated analysis. Figure 2 shows results from the Liang (left two panels) and the Ziska (right two panels). We decided to use the Liang emissions because they were consistently higher at a similar percentage compared to VSLS, whereas the Ziska emissions are different for each gas at a greater magnitude.

*3) What would cause land sources of CHBr3 and CH2Br2? In the introduction, only marine sources are discussed, but later the reader is confronted with the land tagged tracer and its contribution to the observed mixing ratio.*

This comment is common to Reviewer 1, which we address above.

*4) The discussion of the model evaluation (section 4.1) needs to be improved. How large are the relative deviations between model and observations. If the bias is mostly a result of the emissions used, than the relative differences should stay constant with height. If however, the relative differences increase or decrease with height this would indicate errors introduced by the transport scheme of the model. Even tough, there are six panels used to discuss the comparison such conclusions are currently not possible.*

This comment is common to Reviewer 1, which we address above.

*5) Please provide the model resolution. At the moment only the resolution of the meteo-rological input data is given. Is this the same as the model resolution and the resolution of the output data? How would this quite coarse resolution (2 × 2.5) impact the results? In particular, how would this impact the model-based analysis of the observations?*

We have now included the resolution, which is the same as the driving meteorological data. The model resolution would potentially impact some of the model results, particularly for tagged tracers in coastal regions around the smaller islands. We now include the role of model resolution in our discussion of the CAST and CONTRAST age profiles.

*6) Please improve description and discussion of Figures 9, 10 and 11. It is difficult to understand what has been done and why some of the statements are made. See also detailed comments further below.*

Agreed. We have clarified the discussion of these Figures.

**Minor comments**

*Page 5, line 4-12. Please explain Figure 2. Are the tagged tracer regions shown or are the tracer regions combined with the emission scenario shown? How do you end up with 20 tagged tracers? Seven for CHBr3 and seven for CH2Br2 and the rest for total and background?*

We show tagged regions with emission flux in Figure 2. We have better described the Figure in the caption and main text. We describe the 20 tracers in the tagged model methodology. In addition, to describe the (now) three ocean tracers:

FROM
"We assign individual tracers to major islands within the study domain, including Guam (13.5N, 144.8 E), Chuuk (7.5 N, 151.8 E), Palau (7.4 N, 134.5E) and Manus (2.1S, 147.4E). We assume these island land masses account for 100% of a grid box

irrespective of whether their area fills the grid box. We have a total of 18 tagged tracers, evenly split between CHBr3 and CH2Br2 including a total tracer and a background tracer"

TO
"In addition to the ocean tracers, we assign individual tracers to major islands within the study domain: Guam (13.5N, 144.8 E), Chuuk (7.5 N, 151.8 E), Palau (7.4 N, 134.5E), and Manus (2.1S, 147.4E). We assume these islands account for 100% of a grid box irrespective of whether their area fills the grid box. We have a total of 18 tagged tracers, consisting of the total, background, three ocean tagged tracers, and four island tagged tracers for CHBr3 and CH2Br2."

*Page 5, line 15. Please explain what 'de-seasonalized monthly means'? Are you using annual means? Or interannual anomalies plus mean values?*

Deseasonalised monthly means are monthly means with this seasonal cycle removed. We use the same emissions are used throughout the study.
"This emission inventory has global annual totals of 425 GgBryr-1 for CHBr3 and 57 GgBryr-1 for CH2Br2 (Parella 2012). Emissions do not have a diurnal cycle and are emitted over a 30 minutes time resolution over the model period."

*Page 5, line 26. This sentence makes no sense. You use age of air simulations be-cause you have no reliable emission inventory? But then the other half of the analysis is based on one emission inventory? Furthermore, should this sentence suggest that only the bottom-up inventories are unreliable while the top-down are not?*

We agree this is confusing and have modified this statement. Age of air is used to interpret the emission regions independent of the emission scenario. This means that even if the model does not represent observations, the relative influence of emission sources can be investigated over the region. In Section 3.1 describing the tagged model, we acknowledge model evaluation using the Ziska (2013) emission scenario

and that it is also unreliable at reproducing observations from this campaign.

"We use the age of air simulation to understand the role and frequency of rapid convective systems to transport short-lived halogenated compounds to the TTL, independent of the emission inventory. The method uses only knowledge of the distribution of emissions, and not the magnitude, so we investigate the influence of emissions source region with respect to respective $CHBr_3$ and $CH_2Br_2$ atmospheric e-folding lifetimes."

*Page 6, line 21-22. Please explain how the amount of explained variability is estimated. Page 7, line 17. How were those percentage contributions calculated? Transform numbers from Figure 5 into relative numbers and then apply them to the observations? Here and at other places, the methodology is not clear and the reader has to guess what exactly has been done.*

We transform the squared Pearson correlation coefficient from Figure 5 into the variability. We have changed the text to read: "From Pearson correlation coefficients, we find that GEOS-Chem reproduces.."

*Page 8, line 5. I don't understand how the discussion of Figure 9 (which shows age of air as a function of source region but no emissions or mixing ratios) allows such a statement. Or is here information from other earlier analysis used? Same for line 8.*

The discussion of this Figure has now changed due to updated model runs and these lines have been deleted. The text has been clarified to relate the age of air to lifetime of $CHBr_3$ and $CH_2Br_2$, with comparisons to mixing ratios being included in discussion of Figure 10 (was Figure 11).

*Page 8, line 12. The text says that 53% of what reaches the TTL comes from the open ocean? From other parts of the manuscript, I had the impression that the large majority comes from the open ocean? Please clarify.*

These values have now changed due to correcting an error in the age of air model run. In the text, it has been clarified that the percentages relate to the percentage of

emissions from source regions. Currently it reads as:

"We find that 76% (92%) of oceanic emissions reach the TTL within $2\tau$CHBr3 ($3\tau$CHBr3), with 64% (88%) of open ocean emissions and 9% (50%) of coastal emissions reaching the TTL within the same time frame."

*Figure 9: Comparing the lines for ocean, open ocean and coastal ocean, I wonder if the coastal and open ocean together should give the ocean age of air? However, the total ocean (blue line) shows the youngest age of all. Please clarify.*

I have checked this in the model by summing the open and coastal tracers and they are equal to the total ocean (Figure 3).

**Review report 3 (Q. Liang)**

*1. The use of Liang et al. (2010) emissions. I don't understand why there were emissions over the land, as the Liang et al. emissions scheme only specifies emissions from Open Ocean and coastal regions. While the original inventory was derived on 2x2.5 horizontal resolution, I provided a refined emissions inventory on 1x1 degree resolution to the GEOS-Chem group. Could it be possible when the 1x1 degree emissions were regrided to 2x2.5, emissions appeared to occur over the island landmasses as a result of coarse resolution? Whatever the reason was, the use of land tagged emissions tracers for CHBr3 and CH2Br2 and the reference of terrestrial sources of these gases throughout the manuscript, in my view, are not accurate and lead to wrong impression that land could be a source of these oceanic-originated compounds. Second, the Liang et al. (2010) emissions inventory was originally derived for stratospheric bromine budget purposes (therefore without much attention to fine-tuning the surface emissions details, e.g. longitudinally invariable and simple treatment of Open Ocean vs. Coasts), with no observations over the western Pacific to constrain surface emissions in that region. As shown by Hossaini et al. (2013), the Ziska bottom-up inventory is a much*

*more skillful and a more appropriate choice of emissions for the Western Pacific region. Quantifying the relative importance of open ocean emissions vs. coastal sources using the Liang et al. (2010) emissions scheme for the Western Pacific region, which is one of the main focus of this paper, does not provide a credible estimate.*

This is a comment raised by the other reviewers. The land tagged tracer appeared due to the ocean mask used overlapping with the emission fluxes. The land tracer has been incorporated in to coastal tagged tracer as this is the area that has overlapped.

We find that using the Ziska emissions are not better at representing observations over our study region region. Ziska emissions are more inconsistent in representing CHBr3 and CH2Br2 over our study region. They underestimate CHBr3 by around 40% and overestimate CH2Br2 by 35%. Liang is based high for both gases. On the basis on that result, we decided to use Liang emissions. We now include a justification in the revised manuscript.

*2. Page 8, Line 29 – Page 9, Line 9. I have to say I don't see the meaning of the use of modeled profiles of CHBr3 and CH2Br2 by applying a vertical uniform correction of model biases to quantify SGI and PGI. i) First, the estimated injection of PGI based on model corrected profiles is not correct. Why use the model? The model, even after correction, still shows low biases for CHBr3 and CH2Br2 at 10-12 km. In fact, shouldn't the observation-based organic Br be the true PGI value?*

We used this method so we could apply it to our model of CHBr3 and CH2Br2 for future scenarios without observations. It is a simplified method for estimating model PGI at these altitudes.

*ii) While it was not explained in the text, my guess is that the authors use the difference of Br value at the surface and that at the TTL to calculate PGI. This is not a correct approach in my view. As show in Liang et al. (2010), a significant fraction of the inorganic Br produced from CHBr3 and CH2Br2 degradation are removed by largescale precipitation in the lower troposphere and never makes to the UT.*

Agreed. This was a comment raised by another reviewer. It was an oversimplification of the issue and calculation. Consequently, we have decided to remove it from the discussion.

*3. Same as the other reviewers, I also find the use of idealized age of air, in particular the results presented in Figure 11, hard to interpret.*

As described above in response to the other reviewers, we have revised the definition of age of air and the description of the results in the revised manuscript.

Please also note the supplement to this comment:
https://www.atmos-chem-phys-discuss.net/acp-2016-936/acp-2016-936-AC1-supplement.pdf

[Figure]

[Figure]

Original emissions

Elevated coastal emissions

**Fig. 1.** Comparison of tagged tracer regions using original emissions (left) and elevated coastal emissions (right). Box denotes Figure now used in manuscript.

Original Liang emissions

Ziska emissions

**Fig. 2.** Model evaluation from Figure 4 using Liang (left two panels) and Ziska (right two panels) emissions.

[Figure]

**Fig. 3.** Age profile shown in Figure 8 in manuscript with addition of the red line which is calculated ocean from summation of open and coastal tracers.

**Supplement:**

[revised manuscript text omitted]

---

## Referee Report (RR1)

2nd Review of "Quantifying the vertical transport of CHBr3 and CH2Br2 over the Western Pacific" by Robyn Butler et al.

Many improvements have been achieved since the 1st round of review. The authors have properly addressed the specific issues I raised before (Specific comments I, II and III), being the most important the correction of including an independent land tracer for VSL emissions. However, I still find that the description and discussion of the results can be greatly improved, and that the main results found can be written in a more rigorous fashion. Indeed, my general concern during 1st round of review was: *" … and most importantly, I found many of the analysis and discussions given in Sections 4 (Results) and 5 (Discussions and Concluding remarks) very vague and/or requiring stronger evidence that supports them. I suggest being more specific and formal on the explanation of the specific statements highlighted in this review"*; which was not answered nor commented in the response letter.

I found that many of my original issues raised in the review were properly answered in the response letter. But when I read the 2nd draft, those answers were not so clearly described or justified in the manuscript text. For example, the answer letter points out that (P2) *"… e.g., coastal emissions play a larger role at higher altitudes."* or (P7) *"…The total tracer is based on a 12-month spin up from the emission scenario with the 'total ocean' emissions included. The VSLS VMR that is not represented by the tagged tracers will be attributed to 'background' VMR from the spin up file, which represents the VMR from before the campaign. We have clarified this in the method and discussion sections"*. I could not find those (or equivalent) sentences in the manuscript. Please, be sure to include all comments from the response letter into the manuscript.

As I previously mentioned, *"I found the study very interesting and of relevance for ACP"*, but I believe redaction/quality improvements can be achieved before final publication. Those should be oriented to avoid using misleading interpretations if individual sentences are extracted from the paragraph (P1, *"In the absence of reliable ocean emission estimates,…"*; or P5, *"We chose to use Liang et al. (2010) because it has a consistent bias for CHBr3 and CH2Br2."*) and to justify the contributions of other factors/scenarios not explicitly considered in the tagged experiments performed in this work (P7, *"Larger differences in the correlations for CH2Br2 is likely due to differences in the sampled air masses that have originated far upwind."*). Additionally, the manuscript would benefit of a comprehensive discussion on how the Butler et al results relate to previously published papers already in the literature (mainly those performed in the same region of study). Bellow I point out to the main phrases/sentences in the text were I found some ambiguity or confusion that should be improved.

P1, L10: *"In the absence of reliable ocean emission estimates, …"*. If this statement is correct, then none of the tagged simulations would've have any sense. So please be carefull when specifically justifying your work, specially within the abstract and conclusions. You may rather replace "reliable" by "high resolution estimate" or "local estimate".

P1, L9: *"… and by older air masses that originate upwind"*. What do you mean by upwind? That bromine sources are somehow generated above the surface? (See related comment below).

P2, L29: Fernandez et al., 2014 has also provided estimates of PGI and SGI contributions lying within this range.

P3, L5: Specific clarification of using only SG measurements from CONTRAST and CAST should be given, as those campaigns also measured PGs in the UT.

Section 2.1. Provide specific information of the exact dates when the campaign was performed.

Section 2.2. This section only points to Table 2 (which only shows the locations of NOAA stations) and then points to Apendix A-1. I suggest moving the whole NOAA validation section to the Appendix, including Tables 2 and 3, and summarize within a paragraph the main results in the text.

Section 3: When the model description is given, no reference regarding the period of time modelled is provided. The latitude/longitude limits used to define the western pacific region are not defined.

P5,L7: *"Figure 2 shows the magnitude and spatial distribution of our prior emissions of CHBr3 and CH2Br2 (Liang et al., 2010)."* What do you mean by "prior"? You have not modified them into a top-down like approach to adjust the Liang emission to your model. So the emissions are kept constant throughout the whole study. Also, indicate if the Liang inventory includes any "coast-to-ocean" scaling factor that could affect the results and conclusions from your work?

P5,L9: Reported emissions from Liang et al, 2010 are not 396 Gg Br yr$^{-1}$, but 425 Gg Br yr$^{-1}$. Please check.

P5,L14: *"We chose to use Liang et al. (2010) because it has a consistent bias for CHBr3 and CH2Br2."* I can imagine that you can find a better reason for choosing the Liang inventory than this one. Also, is it the ocean tagged version identical to the Liang emissions? (within the WP region)

P6,L9: Rx has not been defined

P7, L11: If the formula for bias computation is included in the main text, then it should be explained. I suggest just moving to the figure caption.

P7, L3: *"Model errors in reproducing the observed seasonal cycle reflect errors in production and loss rates."* How do you compute "production rates" from SGs? Do you mean "errors in VSL sources and loss rates"? All VSL chemical mechanism I am aware of include only decomposition of VSL halocarbons by reaction with OH and hv, so you can compute the loss

rates. But there are not any VSL production rates due to gas-phase reactions, only emission of source gases from the ocean.

P7, L9-10: *"Larger differences in the correlations for CH2Br2 is likely due to differences in the the sampled air masses that have originated far upwind."* Once again, what do you mean by upwind? You should make this explanation clear in the text. Also, please relate the SGs surface analysis to the TransCom-VSL paper (Hossaini et al., 2016) and their findings respect to the global model performance in reproducing VSL SGs in the surface and UT when different emission scenarios are used.

Later in P8, L10-12, the authors cite the TransCom-VSL paper, but they seem to be pointing out to how different models behave differently when different emissions are used, while the TransCom-VSL paper highlights that most global models used were capable of reproducing VSL SG in the TTL independently of the emission inventory used.

P7, L5: I understand the intention of the authors, but I do not see the inverted S shape in the VSL vertical profile in Figure 4. Also in P8, L3 and elsewhere, the inverted S shape is mentioned but is never explained nor justified. Which are the processes producing this observed feature?. (See my comment on Fig. 4 below).

P7, L27: *"Prevailing easterly transport of gases over the region is dominated by the vast area of open ocean sources that appear to weaken the magnitude of spatially limited coastal emissions (Andrews et al., 2016; Pan et al., 2016)."* How can the open ocean "weaken" the coastal emissions?" Please rephrase and explain.

P7, L30: It would be very useful to compare the percentage contribution of coastal emissions to CH2Br2 in the TTL respect to the percentage contribution of CHBr3, and explain it in relation of their predominant sources and lifetimes. Also perform the same comparison for percentage contribution of the open ocean tracers to each species in the TTL.

P7, L33: *"with the remainder originating from emissions prior to the campaigns."*. How are you capable to distinguish that the reminder is from emissions prior to the campaign and not from sources located outside of the tagged tracers?

P8, L2: *"…with contributions from geographical regions immediately outside the study región…"*. How do you know they are "immediately" outside? Have you performed an additional tagged simulation with emissions from the surrounding areas of the WP? How do you relate this statement with your previous comment regarding the emissions originated prior to the campaign (and not outside of the WP)?

P8, L6-9: I found confusing that the "largest" contribution reach a maximum of 28% and that the "reminder" (e.g., 72%) is not referred as the dominant contribution.

P8, L31: *"despite intensive measurements around coastal land masses of the region"*. What do you mean by coastal land masses?

Figure 8: (P8, L15): Why coastal seems not to contribute to the total ocean profile, which seems to be very similar to the open ocean. I would expect the total to be the sum of both coastal and open ocean profiles.

P8, L23-26: I was surprised that the coastal tracer percentage contribution to the TTL was found to be much smaller than the open ocean contribution. Could you provide some insights on this interesting result? Is it because of the assumed source distribution? Is due to the different transport regimes and speed of convection?

P8, L32: Deficiencies could also be due to an incorrect representation of the spatial distribution of VSL sources in the inventory used (you mentioned at other places of the text).

Figure 10 (P8, L34, P9, L3): I do not understand what the intention of including Fig. 10 is nor the analysis performed here. Please describe it in more detail or remove it. Other reviewer also highlighted this issue during the first round of review.

P9, L18-34: I believe that in the discussion a comparison with the results obtained from Navarro et al., 2015 during ATTREX in the same region of study should be given. Also, relate your results to other papers reporting CONTRAST and/or CAST data.

Section 5: There is no discussion at all, only a summary of the results previously presented. So it should only be called "Concluding Remarks".

P10, L4-5: *"…due to advection of air masses convected from areas outside the study region…"*. Is this contribution dependent on the strength of convection or on the large scale ascent?

P10, L9: *"…are dominated by sources from before the campaign."* Here in the conclusion the "reminder" contribution seems to be the dominant source. Once again, the authors need to explain how the contribution from "before" the campaign and from "outside" the region are recognized and distinguished.

Figure 4: You should use a,b,c,d,e,f labels for each independent panel. In panels a.b (top row) I suggest using filled colored boxes for model output and empty colored boxes for campaign data. Also explain that the model and observations data corresponding to the same altitude interval are slightly shifted in the vertical axes for each bin.

Figure 8: Indicate in the figure caption what the vertical dashed lines indicate.

Figure A-2: The sigma errors are denoted by vertical (not horizontal) lines.

Appendix A: The information given in the last paragraph could be moved as a summary of the NOAA validation into the main text. Table 3 should be in the appendix.

P29, L4-5: *"This variability will represent the large variability of convective events over the region, as well as the aforementioned errors in model emissions."* This sentence is confusing and out of context in the appendix. Note that the appendix should be read as an independent portion of the work.

---

## Author Response (AR2)

**Response to second round of reviewer comments of Quantifying the vertical transport of $CHBr_3$ and $CH_2Br_2$ over the Western Pacific by Butler et al**

We thank again the reviewer for these further comments. We have addressed each reviewer comment (denoted by italics) and changed the manuscript where appropriate. We apologize for the delay in our responses.

*P1, L10: "In the absence of reliable ocean emission estimates, …". If this statement is correct, then none of the tagged simulations would've have any sense. So please be careful when specifically justifying your work, specially within the abstract and conclusions. You may rather replace "reliable" by "high resolution estimate" or "local estimate".*

We revised this statement to reflect this comment "In the absence of local ocean emission estimates…"

*P1, L9: "… and by older air masses that originate upwind". What do you mean by upwind? That bromine sources are somehow generated above the surface? (See related comment below).*

They originate from the region outside of our study region. We have clarified this in the manuscript by stating "…but it is still dominated by emissions from the open ocean and by older air masses that originate outside of our study region."

*P2, L29: Fernandez et al., 2014 has also provided estimates of PGI and SGI contributions lying within this range.*

We have now included this reference.

*P3, L5: Specific clarification of using only SG measurements from CONTRAST and CAST should be given, as those campaigns also measured PGs in the UT.*

We have now changed this statement to "We use source gas data from two coordinated aircraft campaigns…"

*Section 2.1. Provide specific information of the exact dates when the campaign was performed.*

We have added dates to the opening sentence in Section 2.1: "We use CHBr3 and CH2Br2 mole fractions from the CAST and CONTRAST aircraft campaigns, running from 18/01/2014-28/02/2014"

*Section 2.2. This section only points to Table 2 (which only shows the locations of NOAA stations) and then points to Appendix A-1. I suggest moving the whole NOAA validation section to the Appendix, including Tables 2 and 3, and summarize within a paragraph the main results in the text.*

We have now moved Table 3 to the appendix. In Section 4.1 we have left a short paragraph summarising results from the NOAA evaluation.

*Section 3: When the model description is given, no reference regarding the period of time modelled is provided. The latitude/longitude limits used to define the western pacific region are not defined.*

We thank the reviewer for spotting this oversight. We have modified the text accordingly:
 "We initiate the model on 1 January 2014 and run until 1 March 2014.  The Western Pacific region is defined as 120º—170ºE and 30ºN—20ºS.  This encompasses the full region covered by the CAST and CONTRAST measurements." In the model description.

*P5,L7: "Figure 2 shows the magnitude and spatial distribution of our prior emissions of CHBr3 and CH2Br2 (Liang et al., 2010)." What do you mean by "prior"? You have not modified them into a top-down like approach to adjust the Liang emission to your model. So the emissions are kept constant throughout the whole study. Also, indicate if the Liang inventory includes any "coast-to-ocean" scaling factor that could affect the results and conclusions from your work?*

The term 'prior' is a slight misuse of language.  This has been changed to "Figure 2 shows the magnitude and spatial distribution of the $CHBr_3$ and $CH_2Br_2$ emissions (Liang et al., 2010)."

*P5,L9: Reported emissions from Liang et al, 2010 are not 396 Gg Br yr−1, but 425 Gg Br yr−1. Please check.*

We thank the reviewer for spotting this typo.

*P5,L14: "We chose to use Liang et al. (2010) because it has a consistent bias for CHBr3 and $CH_2Br_2$." I can imagine that you can find a better reason for choosing the Liang inventory than this one. Also, is it the ocean tagged version identical to the Liang emissions? (within the WP region)*

The reviewer is right that we could have used a number of arbitrary criteria to select an emission inventory, but we chose Liang because of we found a consistent mean bias of these two gases. This in turn allows us to confidently analyse the ratio of these two gases in a complementary paper (Feng et al, https://doi.org/10.5194/acp-2017-949). We confirm that the ocean tagged region represent the Liang emissions.  For VSL sources, we are limited from the number that we could choose.

*P6,L9: Rx has not been defined*

We thank the reviewer to spotting this error. We have changed the text to read "Fractional contributions of tracers are calculated based on relative ratio of each tracer within a grid box (Rx)…"

*P7, L11: If the formula for bias computation is included in the main text, then it should be explained. I suggest just moving to the figure caption.*

Agreed. The bias calculation is now in the caption of Figure 4.

*P7, L3: "Model errors in reproducing the observed seasonal cycle reflect errors in production and loss rates." How do you compute "production rates" from SGs? Do you mean "errors in VSL sources and loss rates"? All VSL chemical mechanism I am aware of include only decomposition of VSL halocarbons by reaction with OH and hv, so you can compute the loss rates. But there are not any VSL production rates due to gas-phase reactions, only emission of source gases from the ocean.*

Agreed. We have this changed this to "emission and loss rates."

*P7, L9-10: "Larger differences in the correlations for CH2Br2 is likely due to differences in the sampled air masses that have originated far upwind." Once again, what do you mean by upwind? You should make this explanation clear in the text. Also, please relate the SGs surface analysis to the TransCom-VSL paper (Hossaini et al., 2016) and their findings respect to the global model performance in reproducing VSL SGs in the surface and UT when different emission scenarios are used.*

We agree this is a difficult statement to understand. We have now changed the text to:

A recent model inter-comparison showed that different combinations of models and prior emission inventories resulted in large variations of surface model concentrations at station sites [Hossaini et al, 2016]. This study that the Ziska inventory was most consistent with observations but not for all models. Model agreement with UT observations was generally better but still showed large inter-model variations particularly over the Western Pacific where there is a strong convective transport.

*Later in P8, L10-12, the authors cite the TransCom-VSL paper, but they seem to be pointing out to how different models behave differently when different emissions are used, while the TransCom-VSL paper highlights that most global models used were capable of reproducing VSL SG in the TTL independently of the emission inventory used.*

Hoissani et al report that
"Overall, model–measurement agreement of CHBr3 in the TTL is poorer during the ATTREX campaigns, with most models exhibiting a low bias between 14 and 16 km altitude. MOZART and UKCA simulations (which prefer the Liang CHBr3 inventory) exhibit larger mixing ratios in the TTL, though are generally consistent with other models around the tropopause. Most (≥ 70 %) of the models reproduce CHBr3 at the tropopause to within ±1σ of the observed mean and all the models are within the measured range (not shown) during both ATTREX campaigns. Model–measurement CHBr3 correlation is > 0.8 for each ATTREX campaign, showing that again much of the observed variability throughout the CHBr3 profiles is captured."

We agree that transport processes are key in understanding VSL SG in the TTL, but there *is* a role for emission inventories.

*P7, L5: I understand the intention of the authors, but I do not see the inverted S shape in the VSL vertical profile in Figure 4. Also in P8, L3 and elsewhere, the inverted S shape is mentioned but is never explained nor justified. Which are the processes producing this observed feature?. (See my comment on Fig. 4 below).*

We have addressed this comment below when the reviewer raises the issues of better labelling.

*P7, L27: "Prevailing easterly transport of gases over the region is dominated by the vast area of open ocean sources that appear to weaken the magnitude of spatially limited coastal emissions (Andrews et al., 2016; Pan et al., 2016)." How can the open ocean "weaken" the coastal emissions?" Please rephrase and explain.*

We have rephrased this to "Prevailing easterly transport of gases over the region is dominated by the vast area of open ocean sources that appear to dominate the magnitude of spatially limited coastal emissions (Andrews et al., 2016; Pan et al., 2016)."

*P7, L30: It would be very useful to compare the percentage contribution of coastal emissions to CH$_2$Br$_2$ in the TTL respect to the percentage contribution of CHBr$_3$, and explain it in relation*

*of their predominant sources and lifetimes. Also perform the same comparison for percentage contribution of the open ocean tracers to each species in the TTL.*

With respect for the sake of readability, we do not understand how this requested change would add to the paper as it is now, especially given its current length. We already show how much of each gas enters the TTL from individual ocean sources.

*P7, L33: "with the remainder originating from emissions prior to the campaigns.". How are you capable to distinguish that the reminder is from emissions prior to the campaign and not from sources located outside of the tagged tracers?*

We define "Background conditions are representative of atmospheric concentrations before the campaign started as they do not include emissions during the campaign period."

*P8, L2: "…with contributions from geographical regions immediately outside the study región…". How do you know they are "immediately" outside? Have you performed an additional tagged simulation with emissions from the surrounding areas of the WP? How do you relate this statement with your previous comment regarding the emissions originated prior to the campaign (and not outside of the WP)?*

This is not a conclusion we can reach with the experiments we have completed. Consequently, we have changed the statement to "Coastal ocean emissions represent a smaller contribution to $CHBr_3$ at lower altitudes, but increase their influence above 6 km in the CONTRAST data reaching a maximum of 60% of the total CHBr3 tracer in the TTL."

*P8, L6-9: I found confusing that the "largest" contribution reach a maximum of 28% and that the "reminder" (e.g., 72%) is not referred as the dominant contribution.*

We have now emphasized that these are the largest contributions of emissions during the campaign period. "The ocean, in particular the open ocean, represents the largest contributions to total $CH_2Br_2$ of emissions during the campaign period. They typically represent 20% of the total $CH_2Br_2$ and reaching a maximum of 28% in the TTL for the CONTRAST measurements. Maximum contributions of coastal emission sources peak at 15% of total CH2Br2 tracer in the TTL, much less than for CHBr3. The remaining contributions (72%) are representative of the emissions prior to the campaign period."

*P8, L31: "despite intensive measurements around coastal land masses of the region". What do you mean by coastal land masses?*

We mean islands. We have revised this statement "despite intensive measurements around coastal areas of the region".

*Figure 8: (P8, L15): Why coastal seems not to contribute to the total ocean profile, which seems to be very similar to the open ocean. I would expect the total to be the sum of both coastal and open ocean profiles.*

This is representative of the relative strength of each of the emission source regions. The weak emission source region of the coastal ocean will not contribute to the age profile. Due to the smaller area coverage of the open ocean, it will be weakened compared to the total ocean age profile.

*P8, L23-26: I was surprised that the coastal tracer percentage contribution to the TTL was found to be much smaller than the open ocean contribution. Could you provide some insights on this interesting result? Is it because of the assumed source distribution? Is due to the different transport regimes and speed of convection?*

This is in relation to the weak strength of coastal ocean emissions. Coastal ocean sources have a relatively smaller source distribution compared to open and total ocean, likely to lead it to have weaker strength in being transported to the TTL.

*P8, L32: Deficiencies could also be due to an incorrect representation of the spatial distribution of VSL sources in the inventory used (you mentioned at other places of the text).*

The age of air calculation is based on bathymetry data and not VSL sources. This is high resolution data that has been averaged over the model resolution, this model resolution will therefore be the dominant cause of mismatched results.

*Figure 10 (P8, L34, P9, L3): I do not understand what the intention of including Fig. 10 is nor the analysis performed here. Please describe it in more detail or remove it. Other reviewer also highlighted this issue during the first round of review.*

We have clarified this text in response to this comment by emphasizing our key points that a) although CH3Br values appear to be insensitive to age they are in fact the superposition of slow ascension of higher coastal emissions and the faster ascension of lower open ocean emissions. We believe that is a useful observation to include in the paper.

"Figure 10 shows mixing ratios of $CHBr_3$ decreasing with altitude, but remaining fairly constant with increasing age within each altitude range. We find that $CHBr_3$ values are determined mainly by younger air masses from the open ocean and older air masses by coastal emissions (Figure \ref{fig:freq_db_age}). Coastal emissions are associated with the highest surface emissions but they also subjected to slow ascent rates and consequently greater photochemical losses. In contrast, open ocean emissions are lower than coastal emissions but are convected more rapidly and subject to less chemical loss. Consequently, $CHBr_3$ appears to be insensitive to age but is in fact a superposition of young and old airmasses from different origins. From our analysis, we found that $CHBr_3$ values are determined mainly by younger air masses from the open ocean (Figure 8). Within the TTL, higher median mole fractions are associated with the highest model convective mass flux in each age bin. The peak frequency for the mean age of air in the TTL is 48–72 days, corresponding to $3\tau_{CHBr_3}$ and median values of 0.5 pptv $CHBr_3$ from oceanic emission sources, and 0.6 pptv in high convective systems. However, less than 0.5% (2%) of air being transported to the TTL within 24–48 (48–72) days of emission are associated with high convection events. Weaker, mean convection plays an important role in more consistently transporting large mole fractions to the free troposphere that is then transported more slowly to the TTL. "

*P9, L18-34: I believe that in the discussion a comparison with the results obtained from Navarro et al., 2015 during ATTREX in the same region of study should be given. Also, relate your results to other papers reporting CONTRAST and/or CAST data.*

This is already included in the revised manuscript. "This is consistent with \cite{Navarro2015} who estimate VSLS contribution over the Pacific from observations in 2013 and 2014. It estimates 3.27$\pm$0.47~pptv of bromine from CHBr$_3$, CH$_2$Br$_2$ and other minor VSLS sources at the tropopause level (17~km)."

*Section 5: There is no discussion at all, only a summary of the results previously presented. So it should only be called "Concluding Remarks".*

Point well taken. Section 5 is now titled "Concluding Remarks".

*P10, L4-5: "…due to advection of air masses convected from areas outside the study region…". Is this contribution dependent on the strength of convection or on the large scale ascent?*

"Coastal ocean sources typically contribute 20\% to total atmospheric CHBr$_3$ but reach a maximum of 60\% in the TTL due to advection of air masses convected from areas outside the study region due to large-scale ascent."

*P10, L9: "…are dominated by sources from before the campaign." Here in the conclusion the "reminder" contribution seems to be the dominant source. Once again, the authors need to explain how the contribution from "before" the campaign and from "outside" the region are recognized and distinguished.*

Agreed. "are dominated by sources emitted outside the study period." Our tagged approach that quantifies this source contribution is described in section 3.1.

*Figure 4: You should use a,b,c,d,e,f labels for each independent panel. In panels a.b (top row) I suggest using filled colored boxes for model output and empty colored boxes for campaign data. Also explain that the model and observations data corresponding to the same altitude interval are slightly shifted in the vertical axes for each bin.*

We have now labelled each independent panel and used coloured boxes for the model panels. We have also added a mean vertical profile show the backwards 'S' profile shape we describe in the main paper.

*Figure 8: Indicate in the figure caption what the vertical dashed lines indicate.*

I have added the meaning to the figure caption: "One e-folding lifetime of CHBr3 of 24 days (vertical dashed line) and CH2Br2 of 123 days (vertical dotted line) are indicated."

*Figure A-2: The sigma errors are denoted by vertical (not horizontal) lines.*

Thank you for spotting the error. This has been corrected.

*Appendix A: The information given in the last paragraph could be moved as a summary of the NOAA validation into the main text. Table 3 should be in the appendix.*

Table 3 has been moved in to the appendix.

*P29, L4-5: "This variability will represent the large variability of convective events over the region, as well as the aforementioned errors in model emissions." This sentence is confusing and out of context in the appendix. Note that the appendix should be read as an independent portion of the work.*

This sentence is in the appendix at the request of a previous review. But we have since decided to delete it.

---

## Author Response (AR3)

We thank the editor for providing LaTeX tips regarding subscripts. We have edited the manuscript following his suggestions.